



Hydrology and
Earth System
Sciences

# A meteorological–hydrological regional ensemble forecast for an early-warning system over small Apennine catchments in Central Italy

**Rossella Ferretti**[1,2], **Annalina Lombardi**[1], **Barbara Tomassetti**[1], **Lorenzo Sangelantoni**[1], **Valentina Colaiuda**[1], **Vincenzo Mazzarella**[1], **Ida Maiello**[3,1], **Marco Verdecchia**[2], **and Gianluca Redaelli**[1,2]

[1]CETEMPS, University of L'Aquila, L'Aquila, Italy
[2]Department of Physical and Chemical Sciences, University of L'Aquila, L'Aquila, Italy
[3]Centro Funzionale d'Abruzzo, L'Aquila, Italy

**Correspondence:** Rossella Ferretti (rossella.ferretti@aquila.infn.it)

**Abstract.** The weather forecasts for precipitation have considerably improved in recent years thanks to the increase of computational power. This allows for the use of both a higher spatial resolution and the parameterization schemes specifically developed for representing sub-grid scale physical processes at high resolution. However, precipitation estimation is still affected by errors that can impact the response of hydrological models. To the aim of improving the hydrological forecast and the characterization of related uncertainties, a regional-scale meteorological–hydrological ensemble is presented. The uncertainties in the precipitation forecast and how they propagate in the hydrological model are also investigated. A meteorological–hydrological offline coupled ensemble is built to forecast events in a complex-orography terrain where catchments of different sizes are present. The Best Discharge-based Drainage (BDD; both deterministic and probabilistic) index, is defined with the aim of forecasting hydrological-stress conditions and related uncertainty. In this context, the meteorological–hydrological ensemble forecast is implemented and tested for a severe hydrological event which occurred over Central Italy on 15 November 2017, when a flood hit the Abruzzo region with precipitation reaching $200\,\text{mm}\,(24\,\text{h})^{-1}$ and producing damages with a high impact on social and economic activities. The newly developed meteorological–hydrological ensemble is compared with a high-resolution deterministic forecast and with the observations (rain gauges and radar data) over the same area. The receiver operating characteristic (ROC) statistical indicator shows how skilful the ensemble precipitation forecast is with respect to both rain-gauge- and radar-retrieved precipitation. Moreover, both the deterministic and probabilistic configurations of the BDD index are compared with the alert map issued by Civil Protection Department for the event showing a very good agreement. Finally, the meteorological–hydrological ensemble allows for an estimation of both the predictability of the event a few days in advance and the uncertainty of the flood. Although the modelling framework is implemented on the basins of the Abruzzo region, it is portable and applicable to other areas.

## 1   Introduction

Floods and extreme rainfall are among the major natural hazards in Europe with over 1000 fatalities and an estimated cost of about EUR 5.000 billion TS1 in damages, between 1998 and 2009 alone (European Environment Agency, 2010). Italy is one of the countries most exposed to hydrogeological risk in the Mediterranean basin, with more than 90 % of municipalities affected by flood and landslide risk (ISPRA, 2018). From 2013 to 2017, 67 casualties due to floods have been reported, with 26 casualties in 2018 only (IRPI-CNR, 2019). The Mediterranean basin is characterized by a highly urbanized coast and mountain ridges close to the coast. During the autumn season there is an increase of the energy available for storms (Duffourg and Ducrocq, 2011) because

Please note the remarks at the end of the manuscript.

of large gradients of the meteorological quantities caused by the cool atmosphere and the warm sea favouring heat and moisture fluxes. That is why most of the heavy rainfall and floods occur in autumn in the Mediterranean area (Ferretti et al., 2014; Rebora et al., 2013; Rotunno and Houze, 2007; Rotunno and Ferretti, 2003), causing natural disasters in the region. Recently, decadal observations and modelling experiments highlighted the changes of precipitation distribution, frequency and intensity (Van den Besselaar et al., 2013; Scoccimarro et al., 2015) and how those changes affected the hydrological cycle in terms of an increasing frequency of flood events (Drobinski et al., 2018; Marchi et al., 2010). Specifically, a warmer atmosphere than the one nowadays with a large amount of water vapour may lead to an increase of intense to extreme precipitation events (Trenberth et al., 2003; Willett et al., 2008; Giorgi et al., 2011). In a context of the increasing likelihood for future weather extremes, the availability of an accurate meteorological and hydrological forecast system is essential for improving civil-protection early-warning systems, on which community safety and impact reduction directly depend (Penning-Rowsell et al., 2009; Alfieri et al., 2012). Moreover, because of the complex orography of the Italian regions with many small- to medium-sized steep and densely urbanized coastal catchments, a further reduction of the hydrological response time and an increase of flood risks is expected. Indeed, in the framework of the European Flood Directive 2007/60/CE (EU Flood Directive, 2007), these regions are mainly classified as P3 (highly dangerous) zones. Recent studies (Hally et al., 2015; Demargne et al., 2014; Cloke and Pappenberger, 2009; Davolio et al., 2008; Schaake et al., 2007) are focused on the coupling between meteorological and hydrological models in order to improve the quantitative precipitation forecast (QPF) and to predict the floods with a sufficient outlook. The coupling of the meteorological and hydrological models requires meteorological observed or simulated variables (mainly, but not only, precipitation and temperature) used as forcing fields in hydrological models (Cloke and Pappenberger, 2009; Alfieri et al., 2013; Abaza et al., 2017; Wanders and Wood, 2016; Fan et al., 2015; Sangelantoni et al., 2019). Hence, the quality of hydrological forecasts is largely determined by the quality of atmospheric input (Pappenberger et al., 2005), even if the goodness of the hydrological forecast strongly depends on the verification methodology (Pappenberger et al., 2008; Alfieri et al., 2012). Temporal and spatial scales of the atmospheric forcing have to be calibrated according to the catchment features. In the case of small-sized and mountainous catchments, because of a more responsive hydrology to the precipitation events, the discharge predictions require a very accurate precipitation forecast. An accurate, in space and time, precipitation prediction represents one of the most difficult tasks in numerical weather prediction (NWP), resulting from complex processes ranging from large-scale atmospheric dynamics to clouds microphysics. The use of meteorological models with high spatial resolution improves the

QPF, but the estimation of the exact location and space-time evolution is still a challenge. In addition, their high computational cost limits the length of the forecast time which is often not enough to ensure sufficient lead time for actions. A potential solution consists of ensemble prediction systems (EPSs) which represent one of the areas from which the largest benefits in predictive skill have been obtained in the context of NWP (Buizza et al., 2005; Bauer et al., 2015). Even though EPSs are characterized by a lower resolution with respect to deterministic forecasting, their added value belongs mainly to two aspects (Buizza, 2018): (1) predicting the most likely scenario estimating the related probability of occurrence and (2) temporal consistency. Concerning the first point, it is particularly relevant for extreme-event prediction. More in detail, the analysis of ensemble member distribution allows for providing the most likely event magnitude coupled to an estimation of all potential outcomes, which characterizes forecast uncertainty (ensemble member standard deviation or spread). On the other hand, the portion (i.e. frequency) of ensemble member predicting values exceeding empirical thresholds corresponding to extreme events can be derived. For the second point, an ensemble generally provides smaller fluctuations among two or more successive forecast lead times than the deterministic forecast. This larger "inertia" implicitly arises from considering the distribution of the members instead of an individual member. This aspect is again relevant for coherently tracking the temporal evolution of potentially damaging extreme events.

As already discussed in previous studies, ensemble weather prediction systems at different spatial scales and using different approaches (Marsigli et al., 2005; Vie et al., 2011) hold a large potential for hydrological forecasting (Demargne et al., 2014; Cloke and Pappenberger, 2009; Schaake et al., 2007; HEPEX, 2004). In the last decade, the scientific community paid an increasing amount of attention to study the EPS coupled to hydrological models, with the aim of improving early-warning systems on different spatial scales ranging from global to regional (Addor et al., 2011; Alfieri et al., 2012; McCollor and Stull, 2008; Davolio et al., 2008; Calvetti and Pereira Filho, 2014; Hally et al., 2015; Saleh et al., 2016).

In this context, the possibility of quantifying and estimating forecast uncertainties allows the end users of hydrological models to manage the risk and to decide the actions to be taken with the aim of reducing the possible damages (Hamill et al., 2005; Schaake et al., 2007; Alfieri et al., 2012). Although the main uncertainty characterizing hydrological forecast results from the precipitation input, uncertainty characterizing the hydrological sphere represents another point to be carefully considered. Traditionally, only uncertainty pertaining the weather forecast sphere is accounted for (Cloke and Pappenberger, 2009). In fact, being forced by individual ensemble members, the same probabilistic approach previously discussed could be applied to the hydrological model as well. This would allow for characterizing the range for

all potential hydrological scenarios and also to assess how weather prediction uncertainty propagates into the hydrological model. This coupled probabilistic approach could further foster the level of confidence that may be associated with the forecasts. In this work, the traditional approach is followed. Based on Cloke and Pappenberger (2009), the total uncertainty is probably underestimated because of the lack of the hydrological model uncertainty which can be obtained by perturbing, for example, the geometry of the system or the model parameters.

In this paper a preliminary evaluation of a meteorological–hydrological ensemble forecast chain, developed at the Center of Excellence in Telesensing of Environment and Model Prediction of Severe events (CETEMPS) is presented. The meteorological–hydrological modelling chain consists of connecting the dynamically downscaled Advanced Weather Research and Forecasting model (WRF-ARW) to the CETEMPS Hydrological Model (CHyM; Tomassetti et al., 2005; Coppola et al., 2007; Verdecchia et al., 2008). The WRF regional ensemble is built by using as initial conditions all the members (20) plus the control forecast of the Global Forecast System (GFS) from the National Oceanic and Atmospheric Administration (NOAA). The CHyM ensemble is built by using the WRF regional 20-member ensemble plus the control. To assess the reliability of this operational meteorological–hydrological ensemble, a preliminary study of a heavy-precipitation event is used as a test case. A statistical evaluation of the WRF ensemble mean precipitation is performed by using a receiver operating characteristic (ROC) curve considering rain gauge and radar surface rainfall total (SRT) data as reference products. Moreover, several experiments are performed using a few initialization procedures: (1) CHyM is forced by the WRF ensemble mean parameters (CHyM-WRF-MEAN); (2) CHyM is forced using the 20 WRF regional ensemble members plus the control (CHyM-ENS); (3) CHyM is forced by the deterministic high-resolution WRF (HR); (4) CHyM is initialized using the observations (precipitation and temperature). The results of the ensemble chains will be compared to the results of both: experiments 3 and 4.

The newly developed Best Discharge-based Drainage (BDD; both deterministic and probabilistic) index, which is built to detect catchment segments that are most likely to be stressed by weather extreme events, is used to analyse the results of the ensemble meteorological–hydrological chains in terms of maps and time series.

The novelty of this work consists of applying a coupled probabilistic approach to both the weather and the hydrological ensemble forecasts; for a small catchment in complex orography, this improvement has been recognized to be extremely beneficial for flash flood and landslide prediction (Alfieri et al., 2012).

Two different ensemble meteorological–hydrological configurations are proposed: (1) a pseudo-hydrological ensemble forecast where the hydrological model is forced by the mean precipitation produced by the WRF 21-member ensemble and (2) a CHyM ensemble composed of 21 members initialized using the 21 WRF members. The uncertainty (i.e. ensemble member spread) and probability of extreme events (e.g. frequency of ensemble member predicting values beyond defined thresholds) will be provided for both ensemble configurations by the WRF regional ensemble, but for the second configuration, a contribution by the hydrological component may occur. The analysis of these two configurations will help in understanding how weather forecast uncertainty propagates into the hydrological modelling outputs, representing an added value in the prediction of hazardous weather-related events.

Based on the results of previous studies (Cloke and Pappenberger, 2009; Marsigli et al., 2005), in this work we assume that for the meteorological regional ensemble all the GFS members plus the control are sufficient to represent the meteorological uncertainty. Obviously, a larger ensemble would ensure a larger spread, but unfortunately the 51 members of the ECMWF ensemble are not available for operational use. Moreover, a regional ensemble is computationally costly; therefore we should reduce the ensemble in any case. A deeper analysis of the impact of a larger ensemble will be the topic of a next paper, following the work of Buizza and Palmer (1998), though Jaun et al. (2008) showed that using 10 members only can be sufficient for having benefits from an ensemble approach for flood forecasting.

## 2 Case study

On 13 November 2017 a deep upper-level trough, associated with an intrusion of cold air from the Arctic region, entered the Mediterranean area and advected south-westerly flow over western regions of Central Italy. The surface depression was located over Central Italy, advecting an easterly flow over the Adriatic regions (Fig. 1a), and the thermal front was extended from northern Africa to the southern Abruzzo region. In the following 48 h the upper-level trough developed into a cut-off low (Fig. 1b) over the central Mediterranean Sea, and the axis of the surface depression tilted (becoming in phase with the upper-level one) advecting a north-easterly flow over the Abruzzo region. Hence on 15 November 2017, there was advection of warm air at low levels and cold air at the upper ones over the Adriatic Sea, producing a highly unstable environment. Therefore, the event was characterized by two phases: in the first phase from 13 to 14 November 2017 the thermal front produced rainfall over the southern Marche and northern Abruzzo regions (not discussed in the paper). During the second phase, on 15 November 2017, when the axis of the trough tilted, the precipitation moved southward (Fig. 2b), and the advection of cold air also produced snowfall on the eastern side of the mountain ridges of the Abruzzo region.

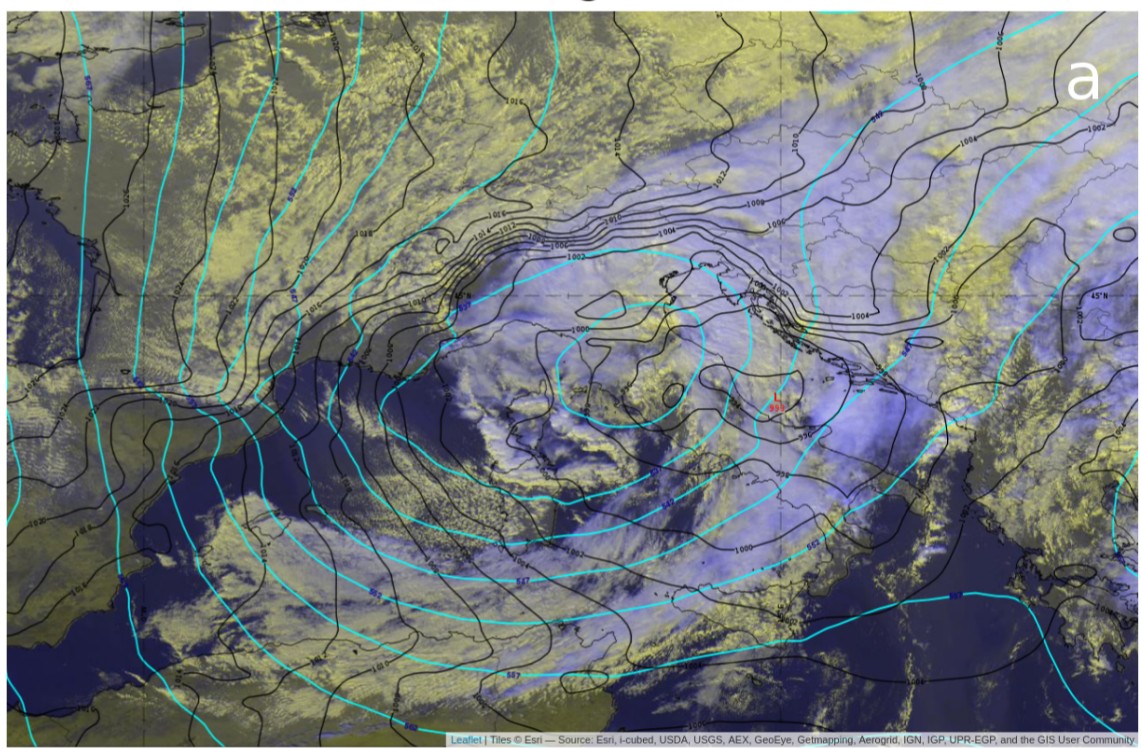

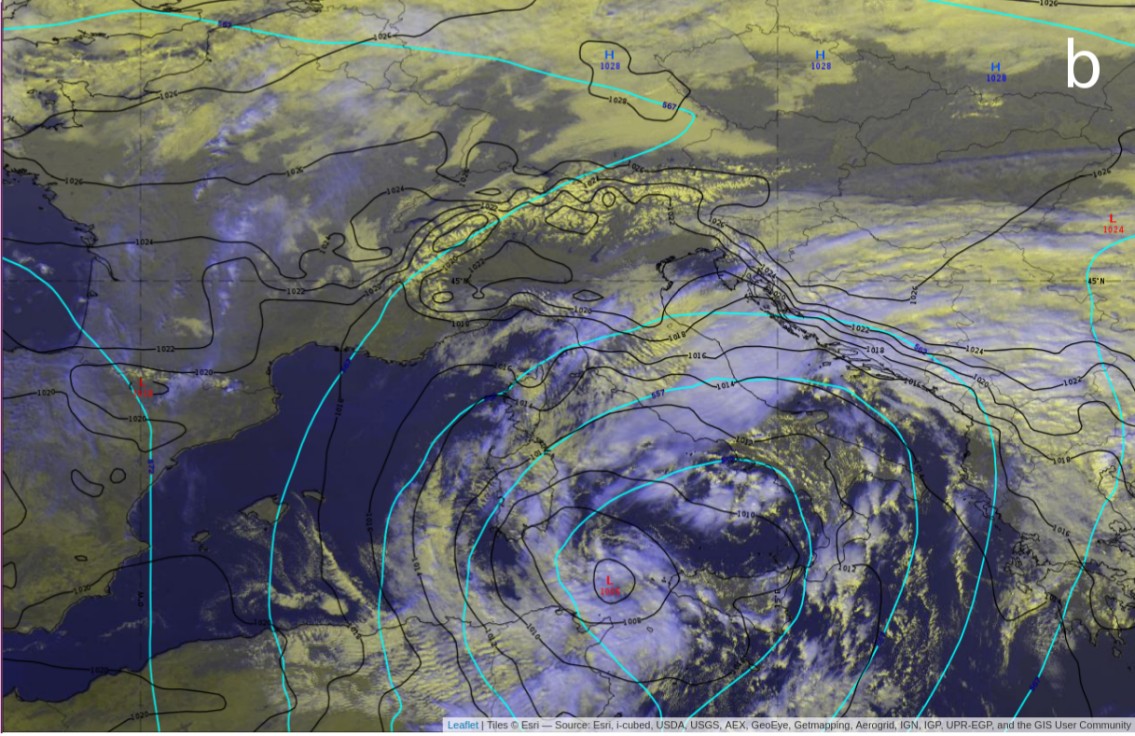

**Figure 1.** ECMWF analysis. Geopotential height at 500 hPa (cyan lines; contour lines of 5 dam) and mean sea level pressure (black lines labelled in hPa; contours lines of 20 hPa) and satellite water vapour (WV) for **(a)** 13 November 2017 at 12:00 UTC and **(b)** 15 November 2017 at 12:00 UTC. The maps have been retrieved from the EUMeTrain (European Organisation for the Exploitation of Meteorological Satellites – EUMETSAT – international training project) ePort archive (http://eumetrain.org/eport.html, last access: 26 September 2016).

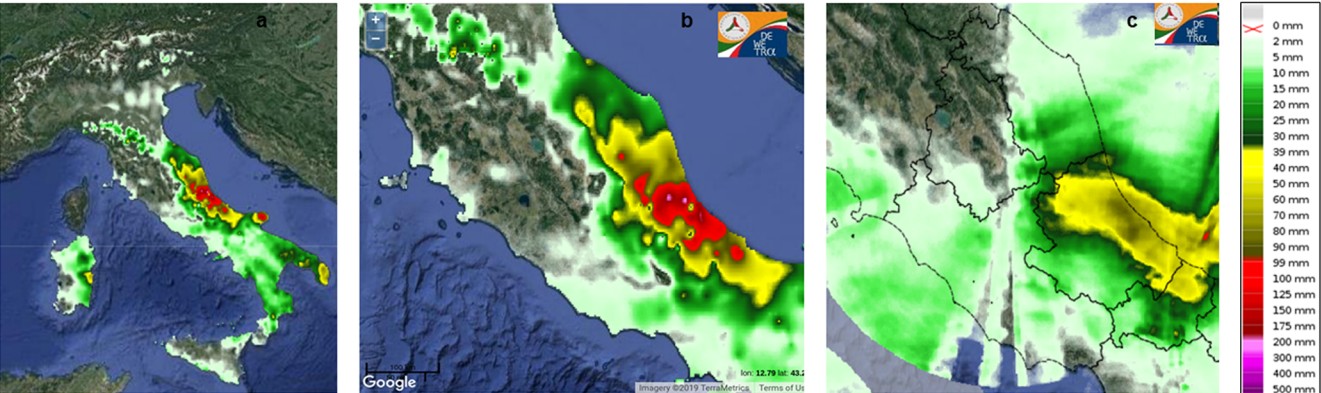

**Figure 2.** Observed accumulated precipitation over 24 h starting on 14 November 2017 at 12:00 UTC: **(a)** over Italy and **(b)** over Central Italy and for **(c)** radar surface rainfall total (SRT). The daily rainfall maps and the radar data are provided by the DEWETRA platform (Italian Civil Protection Department).

Figure 2 shows the accumulated precipitation for 24 h (from 12:00 UTC of 14 November) over Italy; heavy precipitation is found only along the Adriatic regions, with maximum peaks of $200 \, \mathrm{mm} \, (24 \, \mathrm{h})^{-1}$ (Fig. 2a, b) being recorded along the Apennine ridges. The long-lasting rainfall produced effects on the ground over the Adriatic regions, particularly on the Abruzzo region (Fig. 2b) as the alert called by the Civil Protection Department (Dipartimento Protezione Civile; DPC) on the morning of 15 November shows (Fig. 9). This figure shows both the forecast for the alert area and the observation of flooded area as evidenced by the symbols (triangle in the figure) added by the DPC as the event develops. Figure 2c shows the 24 h accumulated radar surface rainfall total (SRT) on the Marche and Abruzzo regions. A similar areal distribution between SRT and the rain gauges is found, but a different amount of precipitation is observed. SRT data and rain gauges will be used for the statistical evaluation of the WRF regional ensemble. The daily rainfall maps and the radar data are provided by the DEWETRA Platform (Italian Civil Protection Department).

## 3   WRF ensemble setup and precipitation forecast

The Advanced Weather Research and Forecasting (WRF-ARW) model is used to build the regional ensemble. WRF-ARW is a non-hydrostatic model with terrain-following vertical coordinates and multiple-nesting capabilities (Skamarock et al., 2008). The configuration for the regional ensemble is the following: one domain covering Italy (Fig. 3, yellow box) with a horizontal resolution of 9 km and 40 unequally spaced vertical levels up to 100 hPa, with higher resolution in the planetary boundary layer. The ensemble is built using all the members (20) and the control forecast (CNTR) from the GFS ensemble. The horizontal resolution of the GFS ensemble system is 1°; these analyses and forecasts are used to produce a dynamically downscaled ensemble fore-

cast at 9 km. Several configurations have been tested, and the following is the one producing the best results in terms of precipitation forecast, at this resolution:

- *radiation* using the rapid radiative transfer model (RRTM; Mlawer et al., 1997) for long-wave and Dudhia scheme (Dudhia, 1989) for short-wave radiative processes;

- *cumulus* using the Kain–Fritsch scheme (Kain, 2004);

- *microphysics* using the Hong and Lim (2006) single-moment bulk scheme, 6-class hydrometeors;

- *boundary layer and turbulence* using the Mellor–Yamada–Janjić (Janjić, 1994) one-dimensional prognostic turbulent kinetic-energy scheme with local vertical mixing; and

- *surface* using the Monin–Obukhov–Janjić surface scheme with the Noah Land Surface Model (Niu et al., 2011).

Moreover, based on the high-resolution (1 km) deterministic forecast operationally performed using WRF over the Abruzzo region at CETEMPS since 2016 (Pichelli et al., 2017), a simulation initialized using the best GFS high-resolution analysis and forecast at 0.25°, updated every 6 h, is performed for this event, and it is used as a benchmark (HR).

All simulations start at 12:00 UTC on 13 November 2017, and they end at 12:00 UTC on 16 November 2017. The boundary conditions are updated every 6 h with the GFS member forecasts for the WRF regional ensemble and every 6 h with the GFS high-resolution deterministic forecast for HR.

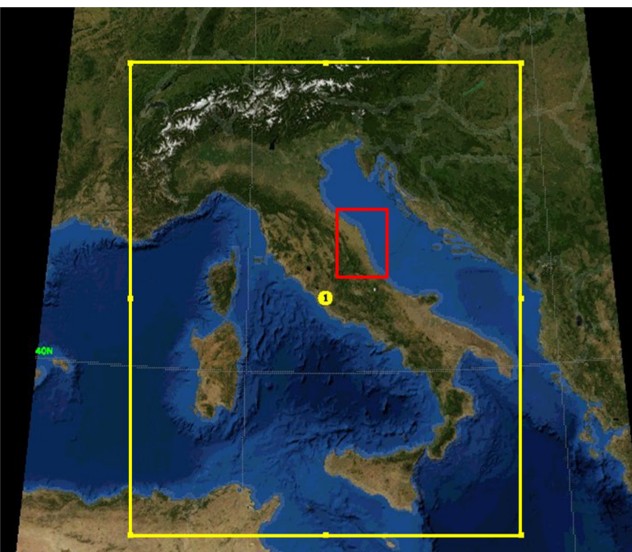

**Figure 3.** Domain for the WRF regional ensemble at 9 km (yellow box). The red box indicates the area considered for the statical evaluation covering the Marche and Abruzzo regions.

**Ensemble member precipitation forecast**

The accumulated precipitation and the associated weather characteristics produced by the 20 WRF members and the control are quite different. An example of the variability obtained for forcing the WRF regional ensemble using the 20 GFS members is discussed by analysing a few of the WRF members (01, 14, 19, 20 and CNTR). Similarly to most of the members, member 01 (Fig. 4b) clearly shows a large area of precipitation over the central Tyrrhenian Sea produced by the surface depression, whereas the area of heavy precipitation at the border between the Marche and Abruzzo regions, reaching $200 \, \mathrm{mm} \, (24 \, \mathrm{h})^{-1}$ (magenta area), is clearly driven by an orographic forcing. Member 14 shows (Fig. 4c) a similar pattern, but a second maximum in southern Abruzzo is found (magenta areas). Similarly to member 14, member 19 produces two maxima of precipitation in the Abruzzo region (Fig. 4d, magenta areas) and a larger cell on the western side of the Apennine ridge with a higher maximum of precipitation (dark-yellow area) than member 14. Member 20 strongly reduces the areal extent of the precipitation for both cells in the Abruzzo region, with respect to members 19 and 14 showing a small area with peaks up to $200 \, \mathrm{mm} \, (24 \, \mathrm{h})^{-1}$ (Fig. 4e magenta area). The CNTR produces the largest cell in southern Abruzzo, with the amount of precipitation reaching values of $200 \, \mathrm{mm} \, (24 \, \mathrm{h})^{-1}$ (Fig. 4a). The cell on the northern side is comparable with the one produced by most of the members.

A qualitative comparison between the WRF member forecast and the observed accumulated precipitation (Fig. 2b) suggests that all members catch the signal of heavy precipitation on the northern side of the Abruzzo region, but all overestimate the areal distribution of the maximum. A good agreement with the observed precipitation is found for member 20 concerning the areal distribution and the maximum precipitation of the cell on the northern side of Abruzzo, though a second cell on the eastern side is missed. In addition, both the member and the control forecasts underestimate the observed heavy rainfall along the Abruzzo coast if compared with the observations (Fig. 2b).

## 3.1 Ensemble precipitation statistics

The information provided by the EPS relies on the analysis of three different statistics derived from the ensemble member distribution:

– ensemble mean from the 20 ensemble members and the control run

– ensemble standard deviation

– probability of the rainfall (or any other meteorological variables) exceeding a given threshold (derived from mapping the ensemble member distribution with respect to the threshold).

To avoid the linear dependency between the standard deviation value and mean ensemble precipitation, the coefficient of variation is computed allowing for an assessment of the precipitation uncertainty independently of the amount of rainfall. For what concerns the threshold in Fig. 5c, it has been arbitrarily defined, and it can be adjusted in the function of regional or local features, season, and the length of the forecast period. Moreover, considering each ensemble member and an equiprobable scenario of future conditions, each member is equally weighted, and the ensemble-based probability can be computed as a binary probability for each member ($P_i$), which is 0 or 1 depending on the occurrence (1) or nonoccurrence (0) of exceeding a given threshold (Schwartz et al., 2010). Therefore, at each time step and for each grid point, the following is computed (Eq. 1):

$$P_i = \begin{cases} 1 & \text{if} \quad \mathrm{var}(i) \geq \text{threshold} \\ 0 & \text{if} \quad \mathrm{var}(i) \leq \text{threshold} \end{cases}$$

$$\text{then} \quad \mathrm{freq} = \frac{\sum_{i=1}^{t_{\mathrm{mem}+1}} P_i}{t_{\mathrm{mem}+1}} \tag{1}$$

where $\mathrm{var}(i)$ is any meteorological variable of the $i$ member and $t_{\mathrm{mem}}$ is the total number of the members plus 1 because the control member is also considered.

The comparison between the 24 h accumulated ensemble mean precipitation at 12:00 UTC on 15 November 2017 computed using all members (Fig. 5a) and the control simulation clearly shows a reduction of the areal extent of the cell in the northern Abruzzo region and a reduction of the precipitation in the southern Abruzzo region with respect to the control run (Fig. 4a). This is an expected result confirming the dampening effect of the ensemble mean if compared to

Accumulated precipitation at 12:00 UTC on 15 Nov 2017

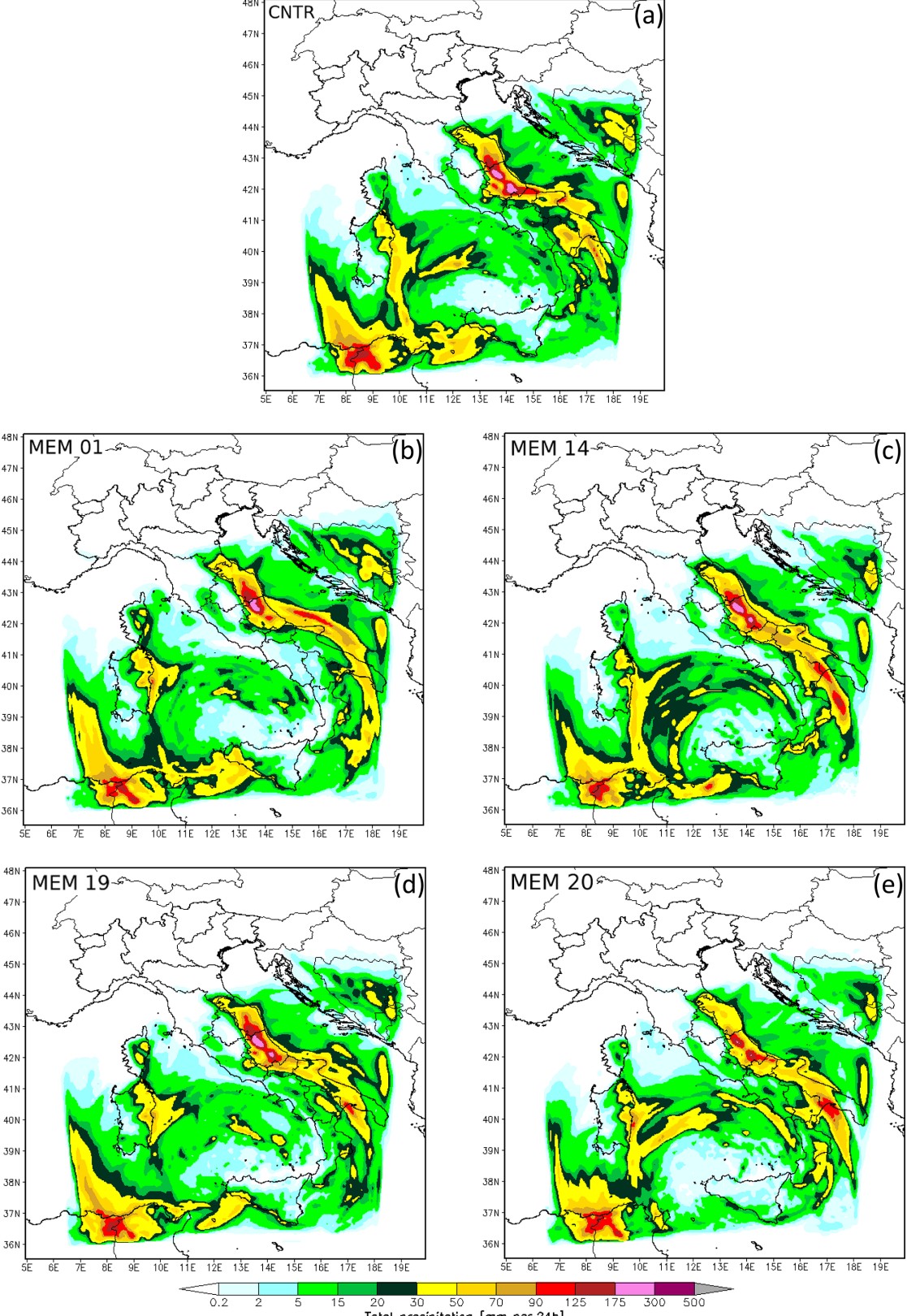

**Figure 4.** WRF accumulated precipitation over 24 h produced by GFS initial conditions for: **(a)** control (CNTR) and members **(b)** MEM 01, **(c)** MEM 14, **(d)** MEM 19 and **(e)** MEM 20.

Ensemble mean precipitation at 12:00 UTC on 15 Nov 2017

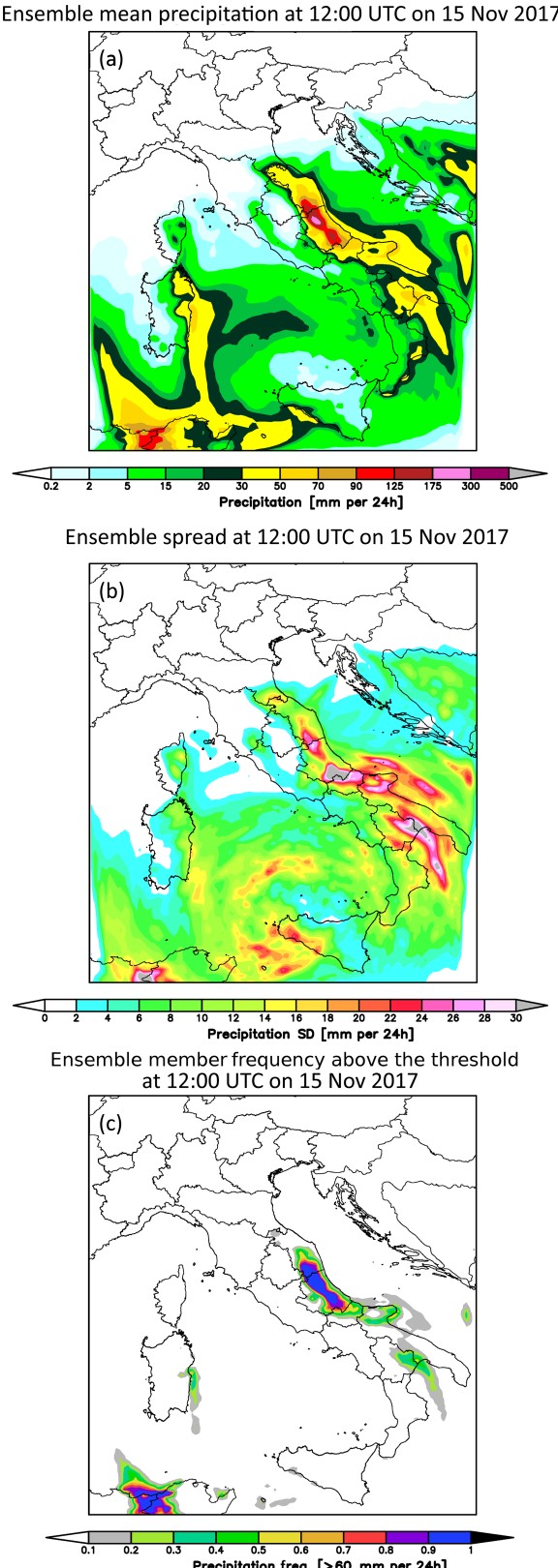

**Figure 5.** Accumulated precipitation of 24 h at 12:00 UTC on 15 November 2017: **(a)** ensemble mean precipitation, **(b)** ensemble spread and **(c)** ensemble probability of precipitation above 60 mm $(24\,h)^{-1}$ produced by the members.

the deterministic simulation at the same resolution. For this event, even though the control simulation takes advantage of the best GFS forecast initial conditions (ICs), it produces a forecast poorer than the one using all members. In fact, the ensemble mean produces a forecast closer to the observations (Fig. 2b) than CNTR, reducing both the areal extent of the cell on the northern side of Abruzzo and the overestimation in southern Abruzzo. However, it is noteworthy that the cell located in southern Abruzzo shows higher values of the spread (Fig. 5b) even if characterized by less accumulated precipitation, suggesting a larger uncertainty in the southern Abruzzo area, for this event. Moreover, in correspondence with the most intense northern Abruzzo cell, there is a small ensemble spread. In fact, the ratio between the standard deviation and the mean is close to 0. This supports the precipitation predicted by the ensemble mean. Besides the ensemble mean value and related uncertainty, the characterization of severe to extreme events represents a focal point. In this regard, the probability of accumulated precipitation above 60 mm in 24 h is shown in Fig. 5c, where almost the whole ensemble (more than 90 % of the members) agrees on predicting precipitation equal to or beyond such a threshold, suggesting a quite confident forecast, though in both the northern and southern Abruzzo regions.

### 3.2 Ensemble precipitation time series

An analysis of the EPS precipitation time series at a few stations located at the foothills of the Apennine (Tossicia, Arsita and Villa Santa Lucia; Fig. 6a) and along the coast (Giulianova, Pescara and Chieti; Fig. 6a) is presented. A comparison is performed among the ensemble members (blue lines), the ensemble mean (cyan line), the deterministic high-resolution (HR, red line) forecast and the observation (black line). The two stations along the coast (Pescara and Giulianova) show the maximum rainfall in the very early morning of 15 November 2017 (Fig. 6b, c, black line) starting from Giulianova. Both the ensemble mean and the HR simulations show a good agreement with the observation in the onset of the rainfall at Giulianova and Pescara, but a large underestimation is found at Giulianova for both simulations (Fig. 6b, red and cyan lines). On the other hand a very good agreement is produced by the ensemble mean at Pescara (Fig. 6c, cyan line) for both the timing and amount of precipitation for the first peak in the early morning of 15 November 2017, but the largest peak of precipitation at the end of the event is misrepresented by both ensemble mean and HR. At both stations HR produces a second peak which was not observed. At Chieti station a fairly good timing is found for both ensemble mean and HR simulations, but an underestimation of the amount of precipitation is found (Fig. 6d, red and cyan lines). For what concerns the stations at the foothills (Tossicia, Arsita and Villa Santa Lucia) a timing disagreement is still found, but the amount of rainfall is better reproduced by both HR and the ensemble mean (Fig. 6e, f, g,

black, red and cyan lines), suggesting a more accurate forecast if the orographic forcing is playing a key role. Finally, all the time series show a variability among ensemble members much smaller than the difference between ensemble mean and observations at the maximum of the rainfall, as for example at Giulianova station on 15 November at 00:00 UTC (Fig. 6b). On the other hand, a larger variability among the members is generally found for small amount of rainfall: for example on 15 November at 12:00 UTC the spread is $15\,\mathrm{mm}\,(3\,\mathrm{h})^{-1}$ with the difference between observation and mean being $5\,\mathrm{mm}\,(3\,\mathrm{h})^{-1}$, whereas at the same station on 15 November at 00:00 UTC the spread is $10\,\mathrm{mm}\,(3\,\mathrm{h})^{-1}$ with the difference between observation and mean precipitation being $23\,\mathrm{mm}\,(3\,\mathrm{h})^{-1}$ (Fig. 6b). This would suggest a large variability of the spread depending on the maximum value of the precipitation; the same is found for the other stations. As expected, a similar behaviour is found for the time series of the GFS ICs precipitation forecast (not shown), which does not include the maximum of observed precipitation among the member variability, but it does show it for other values of the rainfall. Generally, the NWP forecast at low resolution tends to underestimate the rainfall and the maximum of precipitation. As the resolution of the NWP forecast increases, the amount of precipitation tends to get closer to the observed maximum, but still a problem in the exact location and timing of the maximum can be found. Hence, we may expect that this regional ensemble (9 km) underestimates the maximum of the precipitation, and possibly no any member reaches the observed maximum as found for the GFS forecast.

A final comment with respect to the ensemble spread must be made: a reliable ensemble generally shows a large spread, embracing the observation, which is not the case for this event. However, rank histogram evaluation metrics performed at time steps from 12:00 UTC on 15 November 2017 (see Fig. S1 in the Supplement) show a good calibration of the WRF ensemble. In any case, being as this ensemble is an intermediate step to build the hydrological ensemble, which is the final aim of this work, a conclusive assessment on the ensemble reliability can be expressed according to the benefits of the hydrological forecast, if any. This will be discussed in Sect. 5.3.

## 3.3 Ensemble statistical evaluation

With the aim of objectively evaluating the reliability of the WRF regional ensemble, a statistical approach is used. The receiver operating characteristic (ROC; Mason, 1982; Winston, 1988; Buizza and Palmer, 1998), which plots the hit rate against the false alarm rate, is computed to evaluate the ensemble for this event. The 3-hourly precipitation from the ensemble mean and the control (CNTR) and the 3-hourly accumulated precipitation from rain gauges (345 surface stations) and radar surface rainfall total (SRT) data on 15 November at 09:00 and 12:00 UTC are used to build the ROC. The analysis is performed restricting the area to the one where the event

**Table 1.** Area under the ROC curve.

| AUC | 15 November 2017 at 09:00 UTC | 15 November 2017 at 12:00 UTC |
|---|---|---|
| CNTR_OBS | 0.6494 | 0.5796 |
| CNTR_RAD | 0.5598 | 0.6571 |
| ENS_OBS | 0.7360 | 0.7117 |
| ENS_RAD | 0.6379 | 0.7564 |

occurred (Marche and Abruzzo), as shown by the inner box in Fig. 3. Both radar SRT and rain gauge data are interpolated on the model grid, and an inverse-distance-weighting (IDW) conservative method (Jones, 1999) has been used to re-map radar data and rain gauges data at the model resolution. With the aim of evaluating the ability of the WRF regional ensemble to forecast the onset of the precipitation up to the light to moderate one, the following thresholds are chosen to compute the ROC: 1, 3, 5 and 10 mm. The results of the WRF ensemble mean precipitation and the one from CNTR compared with both rain gauge and SRT data are shown in Fig. 7. The steepness of the curve as well as the area under the curve (AUC) are an indication of the skill of the forecast (Storer et al., 2019); therefore we concentrate on these two factors. The WRF regional ensemble shows a high rate of increase of the probability of detection (POD) for thresholds for precipitation up to 10 mm on 15 November at 09:00 UTC, suggesting a good skill if using rain gauges (Fig. 7a, black line). The comparison with radar SRT data, though showing a lower steepness than the previous ones (Fig. 7a, magenta line), still shows an AUC > 0.5 (Table 1), ensuring a skilful forecast. Indeed, the AUC for the WRF ensemble for the first time step is 0.73 if using rain gauges, but it is slightly deteriorating (0.71) at 12:00 UTC. The reverse is found if using radar data, with AUC = 0.63 at the first time step and increasing to 0.75 at the following one, where the steepness of the curves is the same for both ENS_OBS and ENS_RAD up to 10 mm (Fig. 7b, black and magenta lines). On the other hand, CNTR shows a steepness lower than the WRF regional ensemble for both time steps (Fig. 7a, b, red and blue lines) as well as values of the AUC only sightly above 0.5 (Table 1), suggesting an overall good performance of the regional ensemble.

## 4 Hydrological model

The CHyM hydrological model has been developed at CETEMPS by the hydrological group since 2002, and since then it has been running operationally (Tomassetti et al., 2005; Coppola et al., 2007; Verdecchia et al., 2008), producing alert mapping service to support Abruzzo Functional Centre (Centro Funzionale d'Abruzzo; CFA) decisions. The model is based on the kinematic-wave approximation (Lighthill and Whitam, 1955) of the shallow-water wave, and the continuity and momentum conservation equa-

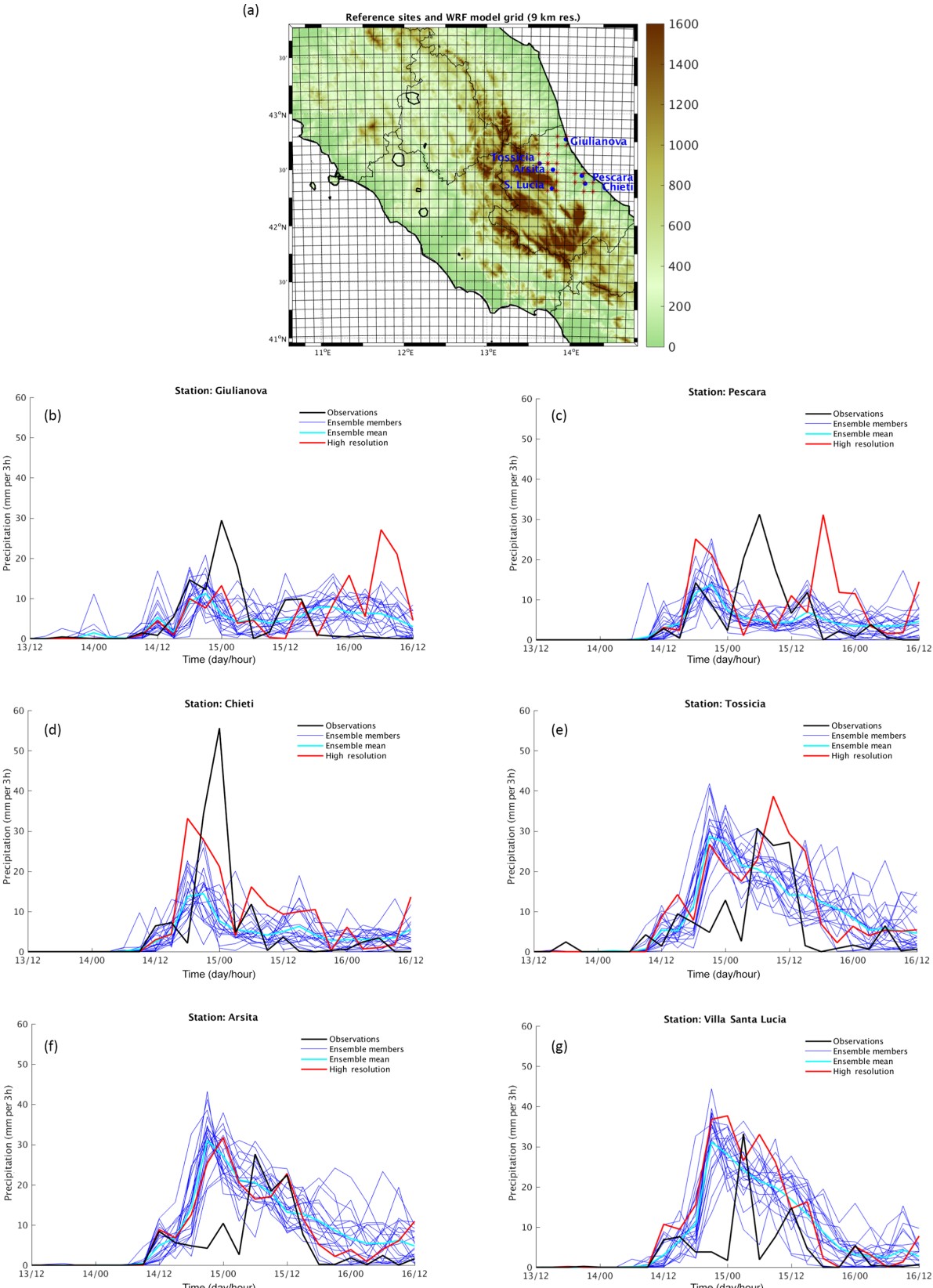

**Figure 6. (a)** Location of the reference stations. Time series (date in November and time of day separated by a forward slash) of precipitation at stations **(b)** Giulianova, **(c)** Pescara, **(d)** Chieti, **(e)** Tossicia, **(f)** Arsita and **(g)** Villa Santa Lucia (abbreviated as S. Lucia in **a**) for the ensemble mean (cyan line), ensemble members (blue lines), HR (red line) and observation (black line).

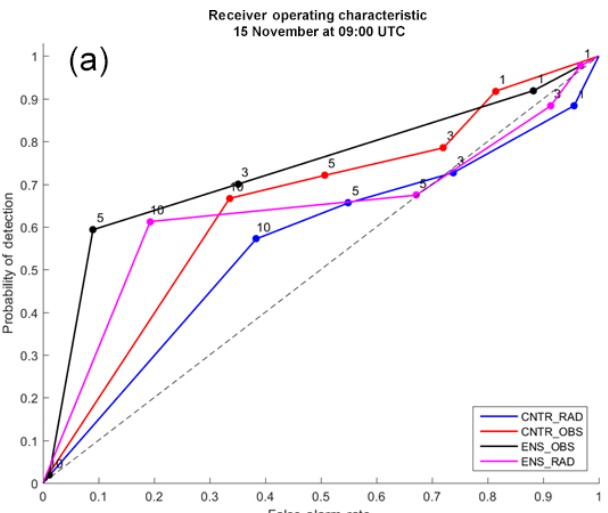
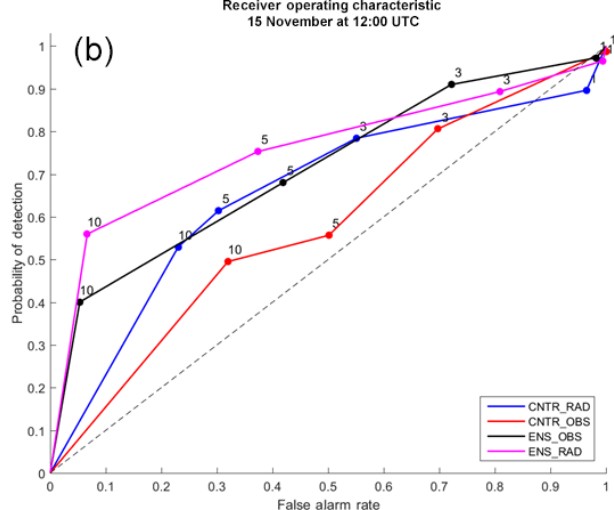

**Figure 7.** Receiver operating characteristic curves using both rain gauge (OBS) and radar (RAD) SRT for the WRF regional ensemble (black and magenta respectively) and CNTR (red and blue respectively) on 15 November at **(a)** 09:00 UTC and **(b)** 12:00 UTC.

tions are used to simulate the surface routing overland and the channel flow. The CHyM model is a distributed grid-based hydrological model reaching a spatial resolution of 300 m; it includes an explicit parameterization of different physical processes contributing to the hydrological cycle. CHyM is forced using different sets of precipitation data which are assimilated and merged in a hierarchical way at each time step. The model can be used for any geographical domain up to the digital-elevation-model (DEM) resolution, and the drainage network is extracted by a sequence of a native algorithm. The interpolation methods for DEM smoothing and meteorological variables spatialization follow an algorithm based on cellular automata (Wolfram, 2002; Coppola et al., 2007). For the Abruzzo region operational activity, CHyM runs at a spatial resolution of 300 m, and it is initialized using observed precipitation and temperature data for a spin-up time of 120 h. The following 48 h forecast is produced using the meteorological-model forecast. For this case study, the same operational configuration is used: CHyM is forced with observed meteorological data until 23:00 UTC on 13 November 2017 and with WRF data for the following 48 h. With the aim of highlighting differences resulting in the hydrological forecasts, a few experiments are performed (Table 2) using the WRF-CHyM chain with different initializations. An experiment is performed by using the WRF ensemble mean (CHyM-WRF-MEAN) and the deterministic high-resolution (CHyM-HR-WRF) simulation, and, finally, a hydrological ensemble (CHyM-ENS) is built. Moreover, two simulations are carried out forcing CHyM with observed data only (CHyM-OBS), which will be used as a control simulation, and the WRF-CNTR output.

### 4.1 BDD index

The analysis of the hydrological forecast is presented in terms of the hydrological-stress index, the Best Discharge-based Drainage (BDD; $mm\,h^{-1}$), which is able to detect catchment segments that are most likely to be stressed by severe weather. The use of a hydrological-stress index is necessary because it is not straightforward to establish a threshold discharge level above which a critical event is to be expected, and such a threshold level should be calculated for each grid point because it depends on the size of the river bed in the selected point. In addition, discharge observations in continuous time series, needed for the calibration, are often missing, especially for small basins. To overcome this problem, we tested a different general definition of an alarm index, and after simulating different case studies occurring in different basins of different sizes, we find that a suitable definition could be the ratio between the maximum value of the predicted discharge, within a given time interval, and the square of the hydraulic radius that is a "measure" of the river cross section for the selected point. The definition of the BDD index also has a simple physical interpretation: it represents the average precipitation available for the runoff drained by each grid element from the upstream basin.

The BDD index is computed at each grid point and time step as the ratio between the flow discharge and the squared hydraulic radius that is the function of the drained area, following Eq. (2):

$$\text{BDD}_{(i,j)}(t) = \frac{Q_{(i,j)}(t)}{R^2_{(i,j)}}, \tag{2}$$

where $Q$ is the discharge estimated by the model, $R$ is the hydraulic radius, and $i$ and $j$ are the grid points. As for many other models the hydraulic radius can be approximated as a

**Table 2.** All CHyM simulations carried out for the case study of 15 November 2017.

| Model | Input | Output | BDD |
|-------|-------|--------|-----|
| CHyM-WRF-MEAN | Mean from all WRF members | CHyM output | BDD mean |
| CHyM-ENS | Each WRF member | CHyM output of 20 members | BDD probability |
| CHyM-WRF-CNTR | WRF CNTR | CHyM output | BDD |
| CHyM-HR-WRF | HR WRF forecast | CHyM output | BDD (from HR) |
| CHyM-OBS | Observation | CHyM output | BDD observation |

linear function of drained area (Singh and Frevert, 2002). In particular, $R = \beta + \gamma D^{\delta}$, where $\beta$, $\gamma$ and $\delta$ are empirical constants to be optimized during the calibration phase. If the area is measured in square kilometres and $R$ is measured in metres, typical values taken from the literature are $\beta$ of 0.0015, $\gamma$ of 0.05 and $\delta$ close to 1 (Singh and Frevert, 2002).

In order to provide suitable and synthetic information for flood alert mapping, it is often useful to plot the map of the maximum value of the BDD index reached within a specified time interval.

Two warning thresholds are defined for the BDD index: medium (orange) and high (red), with a similar meaning to those defined by the civil-protection authorities for the hydrometric height (Thielen et al., 2009). Moreover, as the BDD index is based on the relationship between the computed discharge and the river geometry at each grid point, the defined thresholds are general and applicable over the whole drainage network. According to the indications provided in Alfieri et al. (2012), these characteristics make the index a strong user-oriented instrument, as it is focused on the detection of severity and probability of thresholds exceedances, in order to improve the visualization of early-warning systems.

To assess the flood event occurrences, the hydrometric level threshold exceedances or non-exceedances for the official station network belonging to the Abruzzo Functional Centre is detected for this case study at the station level. A more complete geolocation of registered flood events, outside instrumented fluvial segments, is also inferred from local authorities reports (fire fighters, civil-protection volunteers and police) as shown in Figs. 8 and 9.

A preliminary evaluation of the BDD index is now presented. Figure 10 shows the comparison between two different, though related, normalized physical quantities: time series of the hourly recorded water level and the corresponding BDD time series obtained from the CHyM control simulation (CHyM-OBS), for seven relevant stations highlighted (yellow circle) in Fig. 8. The CHyM-OBS is here assumed as the reference for the BDD threshold definition. In order to ease the comparison, the two quantities (BDD index and water level) are normalized to their respective maximum. This preliminary validation is important because the BDD index obtained from the (CHyM-OBS) will be used in the following analysis as a reference product for the validation of the whole meteorological–hydrological ensemble chain be-

cause of the lack of discharge observations. The comparison is qualitative and mostly focused on the threshold exceedance and maximum timing accordance between the index and the hydrometric level curves. From north to south the following rivers are accounted for: Vomano, Tordino, Saline and Pescara (Fig. 9; the four red triangle-shaped, thin-bounded signs indicate the relevant hydrometers where the red hydrometric threshold has been exceeded). These sensors have been chosen because they are located close to the area where floods or critical hydrological levels have actually been observed (Fig. 8 and related discussion).

The BDD index correctly reproduces the timing and the hydrometric level peak (Fig. 10, red and blue lines respectively) for the first four sensors (Fig. 10a, b, c and d). On the other hand, the Saline sensor shows (Fig. 10e) an uncertainty in the prediction of the peaks of approximately 1 h, which is exactly the time resolution of the series. In addition, for the sensors located in the Pescara catchments, the shape of curve is well predicted, but the hydrometric level is significantly overestimated within a few hours between the two observed peaks (Fig. 10f, g). Furthermore, it has to be considered that discharge for this river is strongly affected by the management of hydroelectric power plants located in the upper part of the basin, and unfortunately no information is available about the management.

Unfortunately, for the Abruzzo region, we are not aware of how the hydroelectric systems are managed. For example, the hydroelectric power plants are located upstream with respect to the areas involved in the event, where precipitation maxima also occurred. This suggests that the effect of the hydroelectric power plants is negligible in this case; the good agreement between the BDD index and water level time series supports this hypothesis.

Therefore, the good performance of the BDD index for this event for the Abruzzo rivers allows for its use in the following analysis.

## 5 Hydrological model results

The experiments presented in Table 2 are now analysed in terms of the BDD index, and the two different ensemble meteorological–hydrological configurations are tested:

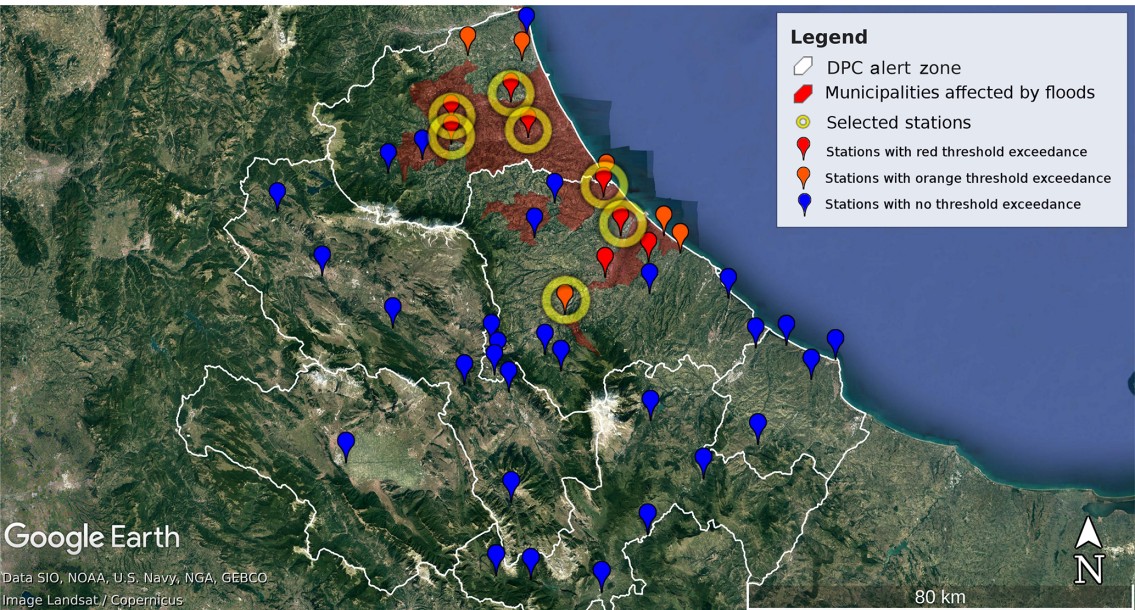

**Figure 8.** A map from Google Earth showing the geolocation of the hydrometric sensors over the Abruzzo region (pinpoints) colour-coded based on the warning threshold exceeded during the event. Yellow-circled sensors represents the subgroup where the analysis at station level is presented. The red areas correspond to the municipalities affected by flood.

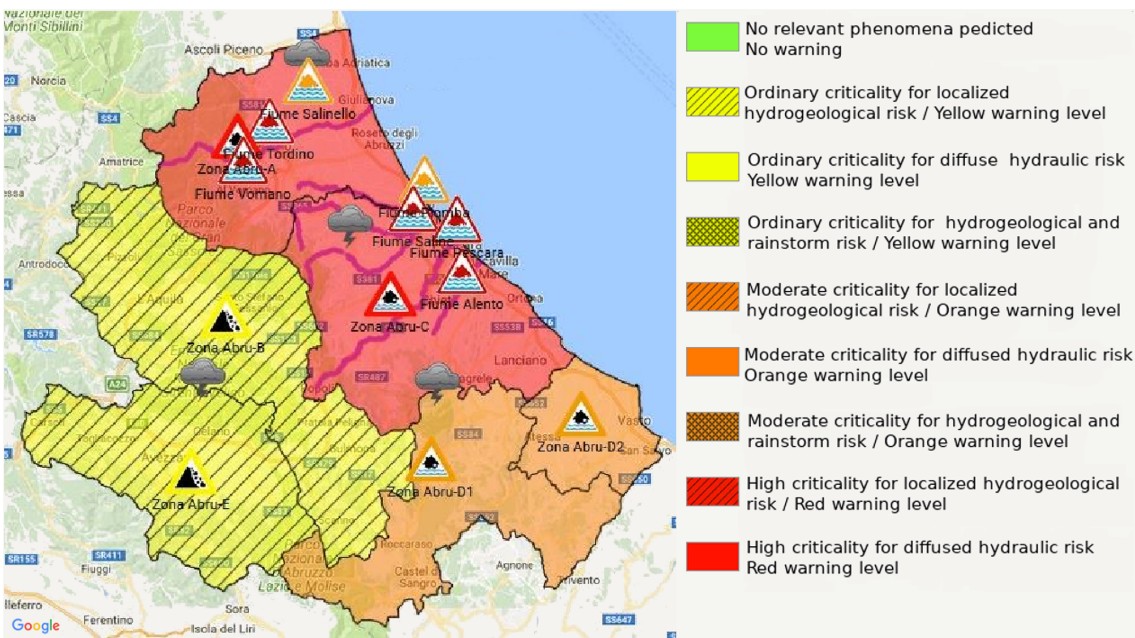

**Figure 9.** Hydrogeological-criticality bulletin issued by the Civil Protection Department on the morning of 15 November 2017, where the Abruzzo region territory is divided into six warning areas (indicated by the prefix "Zona Abru"), coloured according to the included legend. Triangle-shaped, thin-bounded signs are geolocated over relevant hydrometers and coloured according to the colour code explained in Fig. 7, resulting from observed data. Triangle-shaped, thick-bounded signs indicate the forecasted warning level in the reference warning area. Yellow triangles indicate landslide risk at the warning area level, while thunderstorm risk is assigned through a figurative icon. Purple lines on the coastal warning areas highlight river segments where the red threshold has been overpassed. © Google Maps 2019.

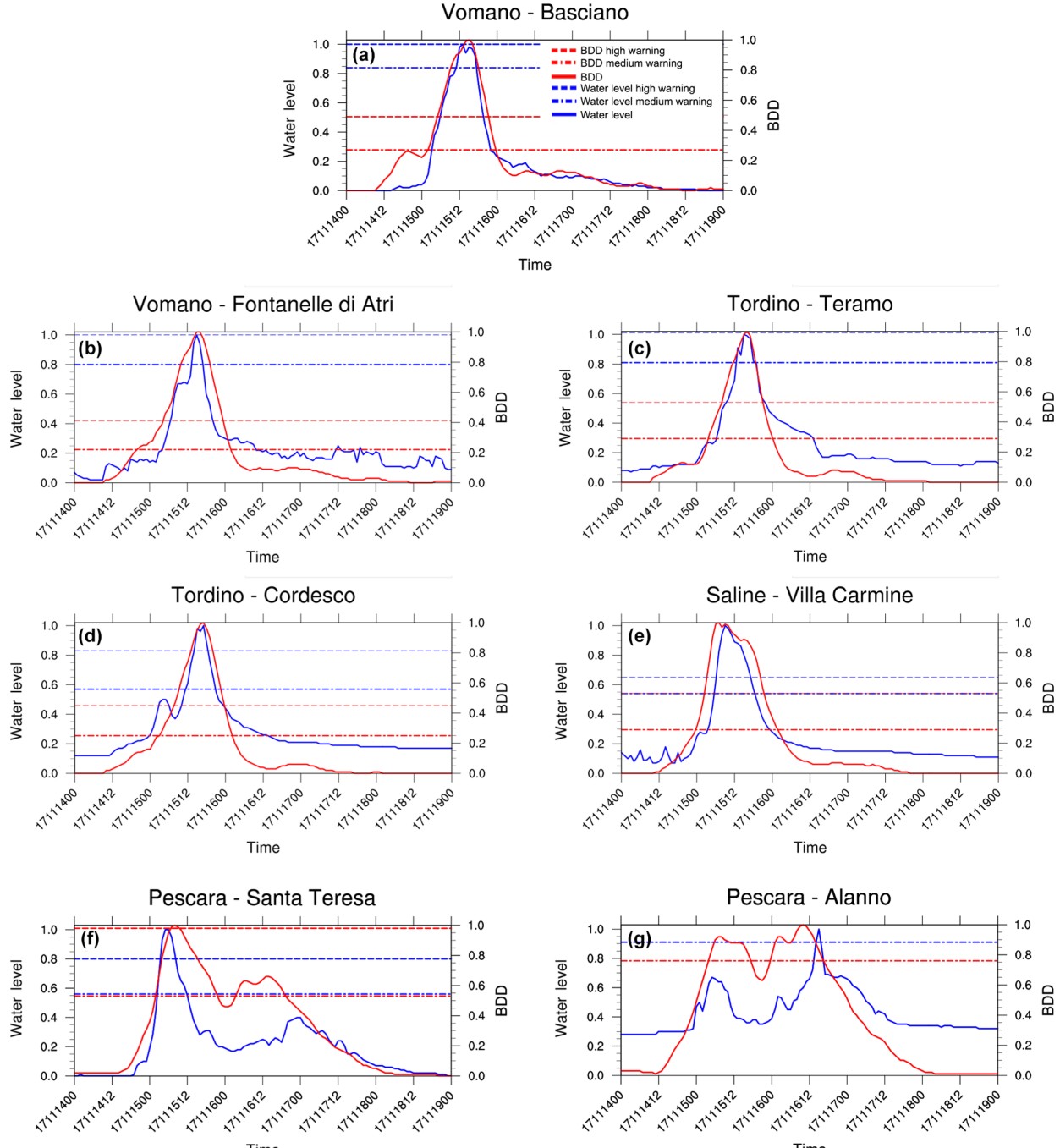

**Figure 10.** Normalized BDD time series (time given in the format of day, month, two-digit year and hour) and water level for the catchments of Vomano, Tordino, Saline and Pescara at stations **(a)** Basciano (Vomano), **(b)** Fontanelle di Atri (Vomano), **(c)** Teramo (Tordino), **(d)** Cordesco (Tordino), **(e)** Villa Carmine (Saline), **(f)** Santa Teresa (Pescara) and **(g)** Alanno (Pescara). The BDD index is represented by red lines, and the observed water level is represented by blue lines.

- a pseudo-hydrological ensemble forecast where the hydrological model is forced by the mean precipitation produced by the WRF 21-member ensemble (CHyM-WRF-MEAN)

- a CHyM ensemble composed of 21 members forced using the 21 WRF members which will be presented in the next paragraphs (CHyM-ENS).

For what concerns the hydro-ensemble, the uncertainty (i.e. ensemble member spread) is provided by the WRF regional ensemble in the first case, whereas a contribution by the hydrological model is expected in the second one.

## 5.1 Pseudo-hydrological ensemble versus hydro-deterministic forecast

At first, the CHyM-WRF-MEAN using the WRF temperature and precipitation ensemble mean is analysed. By means of quantifying the reliability of the ensemble-mean-driven hydrological forecast, 24 h BDD stress index maps obtained considering the CHyM-WRF-MEAN simulation and CHyM-OBS (Fig. 11b, c) are compared, for 15 November 2017. High values (warmer colours) of the BDD index highlight fluvial segments characterized by a high level of hydrological stress, where flooding is most likely to occur. The map is built assigning to each grid point of the drainage network the maximum value of the BDD index calculated according to Eq. (2). Hence, the maps in Fig. 11 represent the worst expected situation from 00:00 to 23:00 UTC on 15 November 2017. A qualitative analysis of the BDD index maps (Fig. 11), obtained with different precipitation scenarios, show good performances of the proposed alarm index by highlighting the areas where major hydrological stress has to be expected. In fact, all the observed flood events shown in Figs. 8 and 9 are correctly predicted by a critical value of the BDD index. The efficiency of the proposed approach seems the same for the main channel as for the small catchments, despite a moderate overestimation of the critical hydrological situation in the southern part of the simulated basin. The comparison between the BDD index by the CHyM-OBS map (Fig. 11a) and Figs. 8 and 9, where actual flooded areas are highlighted along the central and northern Abruzzo drainage network, shows a good spatial coherence. The BDD map by the CHyM-WRF-MEAN precipitation (Fig. 11b) shows a good agreement with the hydrological control run (Fig. 11a), for the main catchments over central northern Abruzzo on the Adriatic side. However, an overestimation along the coast on the southern side of the Abruzzo region and an underestimation on the northern side is found for the small catchments for CHyM-WRF-MEAN. This is probably due to an underestimation of the rainfall along the coast (Fig. 5a) by the WRF regional ensemble as well. In this condition, the main contribution to the hydrological stress (i.e. BDD index) is given by the heavy rainfall produced on the mountains which is able to charge the longest rivers, whereas the shortest streams near the coast do not receive enough precipitation to turn on (warm colour) the BDD index. Furthermore, the hydrological stress is overestimated in the southern part of the domain (Fig. 11b). To further verify the CHyM-WRF-MEAN forecast, a comparison with the deterministic high-resolution forecast (CHyM-HR-WRF) is performed. The BDD index for CHyM-HR-WRF (Fig. 11c) is very similar to the index map resulting from CHyM-OBS (Fig. 11a). In this configu-

ration, the CHyM model is able to capture a higher hydrological stress over the small catchments in the northern coastal area of Abruzzo (Fig. 11c, red colour), which is missed in CHyM-WRF-MEAN (Fig. 11b, light blue). On the contrary, the smallest flooded fluvial segment along the coast, the Calvano stream (on the northern side of Abruzzo but south of the previous one) is not flooded by the deterministic run even if heavy precipitation was recorded. This is caused by precipitation not caught by the meteorological model because it occurs only in a very small area. The stress index over the southern catchments is overestimated, as well. A further verification of the goodness of these results is obtained by comparing the BDD index maps with the alert map issued by the Civil Protection Department of Abruzzo for these rivers (Fig. 9), where a risk of flood was issued. Hence, these results would suggest an overall good performance of both the CHyM driven by the deterministic high-resolution forecast and the one driven by the WRF regional ensemble forecast.

## 5.2 CHyM ensemble

Finally, a CHyM hydrological ensemble forecast is built by performing 21 simulations using the 20 members plus the control of the WRF ensemble as initial conditions. The output of the 21 CHyM members is used to build a probability $BDD_{prob}$ index (Fig. 12), which is computed by the following equation:

$$BDD_{prob} = \frac{N_{BDD}}{N_{ens}}, \tag{3}$$

where $N_{BDD}$ is the number of members for which the BDD index reaches higher values than the alarm thresholds during the 24 h and $N_{ens}$ is the total ensemble members (20, excluding the WRF control run).

To verify this $BDD_{prob}$ index, a comparison with the alert map issued by the Civil Protection Department during the event (Figs. 8 and 9) is presented. The maximum flood probability is found over catchments on the northern Adriatic side (Fig. 12a), where almost all members simulate precipitation peaks, according to CHyM-OBS and CHyM-HR-WRF (Fig. 11a, c). Small catchments stress is not simulated over this area because of a general underestimation of the precipitation amount in the coastal area, probably caused by the lower horizontal resolution used in the meteorological ensemble. An improvement by CHyM-ENS is clearly obtained as shown by the $BDD_{prob}$ index over the southern part of the region (Fig. 12a), where the overestimation of the stress condition detected by CHyM-WRF-MEAN (Fig. 11b) is not found, in line with the real hydrological effect caused by the event in that area. The CHyM ensemble spread is also computed (Fig. 12b), in order to give a complementary information to the $BDD_{prob}$ map. The large value for the spread of the precipitation in the southern Apennine ridge produced by the WRF regional ensemble (Fig. 5c) has implications for the largest catchment of southern Abruzzo: the maximum BDD

BDD index simulation from 00:00 UTC 15 Nov 2017 to 00:00 UTC 16 Nov 2017

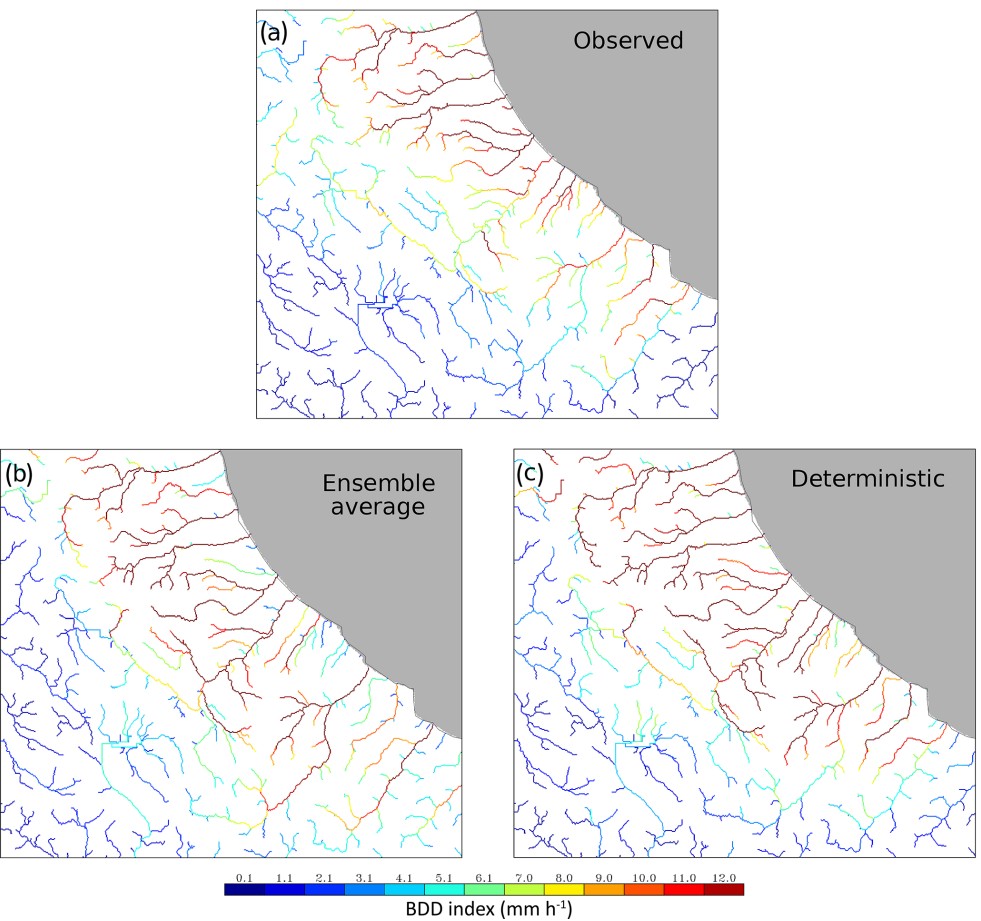

**Figure 11.** The 24 h BDD index computed by CHyM using **(a)** the observed accumulated precipitation, **(b)** the mean precipitation produced by the WRF ensemble and **(c)** the precipitation produced by the deterministic WRF at 1 km (HR).

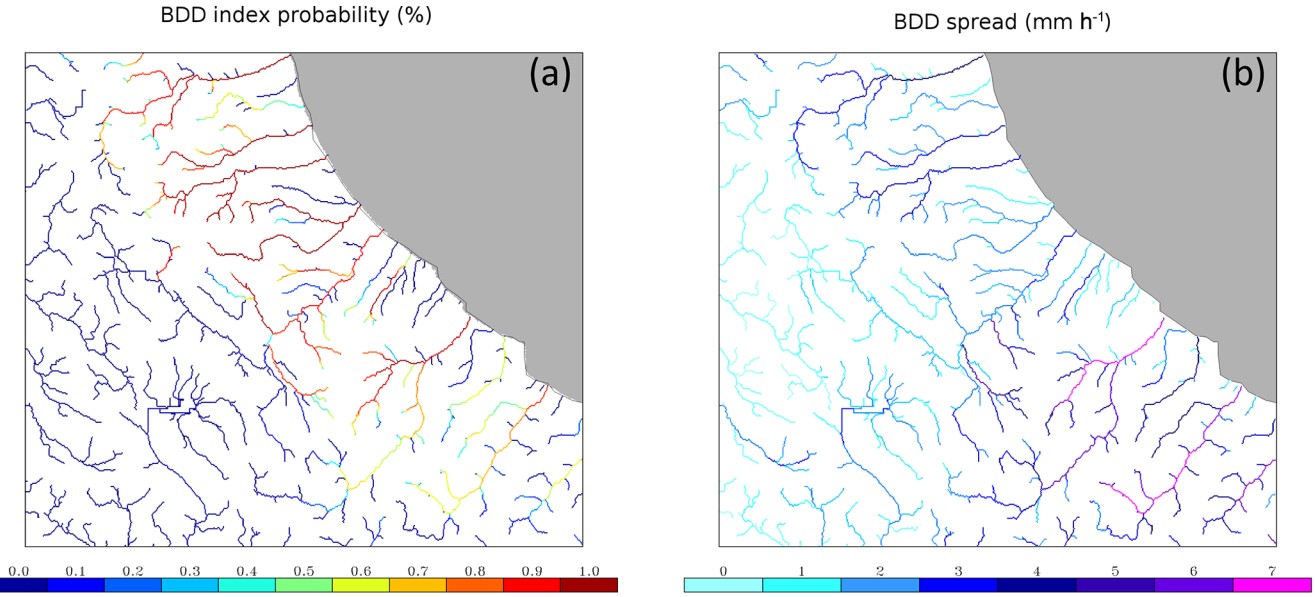

**Figure 12. (a)** The 24 h BDD probability index computed forcing CHyM with the 21 WRF members. **(b)** CHyM ensemble spread.

spread is here obtained for the main catchment, corresponding to the Sangro River, whereas in the northern part of the domain, the spread is smaller, confirming the reliability of the flood forecast. These results suggest a coherent variability between WRF and CHyM.

## 5.3 CHyM time series

With the aim of further evaluating the ability of the hydrological ensemble to correctly reproduce the stress distribution, an analysis of the BDD index time series at a few stations is also presented. CHyM-OBS is compared with the one produced by the pseudo-hydrological ensemble forecast (CHyM-WRF-MEAN), where the hydrological model is forced by the mean precipitation produced by the WRF 21-member ensemble; the CHyM ensemble (CHyM-ENS) composed of 21 members initialized using the 20 WRF members plus the control; and the CHyM-HR-WRF simulation forced using the HR deterministic forecast. In what follows, the BDD index time series are presented for all the CHyM simulations, except for the mean of the 21 CHyM ensemble members because it is similar to the one produced by CHyM-WRF-MEAN. The BDD index time series for the stations along the rivers Vomano, Tordino, Saline and Pescara (from north to south) show (Fig. 13) a BDD spread between 10 and $20 \, \text{mm} \, \text{h}^{-1}$ around the peak. CHyM-WRF-MEAN (Fig. 13, green line) is overestimated at the Pescara River stations, if compared with the BDD time series obtained by CHyM-OBS (Fig. 13, black line). Moreover, it results only in a red BDD threshold exceedance at the Pescara Alanno station, which was actually affected by an orange threshold exceedance. As for the timing, there is a different behaviour between the northern basins and the central ones. For the northward catchments (Vomano and Tordino), the ensemble-modelled peak timing is progressively simulated up to 6 h in advance (Fig. 13a, b, c and d respectively, green and black lines), with respect to the control hydrological simulation. At the Saline-Villa Carmine station, in the central area, the maximum of the BDD index is reproduced with high timing accuracy (Fig. 13e, black and green lines), whereas an approximately 12 h of delay at the Pescara River station (south area) is found (Fig. 13f and g respectively; black and violet lines). The CHyM-HR-WRF input seems not to be affected by the aforementioned time shift. These results would suggest a contribution from the WRF regional ensemble error, caused by the low resolution, in the timing of the maximum peak, as it is found at several stations (Fig. 6) propagating in the CHyM forecasts.

Finally, all the time series show a variability among ensemble members much smaller than the difference between the ensemble mean and observations at the maximum of the rainfall because of the anticipation of the CHyM ensemble mean peak (Fig. 13a, b, c and d). On the other hand, a larger variability among the members is found for Saline time series ($10 \, \text{mm} \, \text{h}^{-1}$) where the timing of the maximum is the same

for both CHyM-OBS and CHyM-WRF-MEAN and the difference between the two is very small ($3 \, \text{mm} \, \text{h}^{-1}$; Fig. 13e). If the time lag would be set to 0 by hypothetically shifting the CHyM-WRF-MEAN maximum at the right time for all time series, we would have the same variability of Saline at all the other stations, except for the Vomano Basciano station. Although the analysed time series (WRF and CHyM) are for different physical quantities, a remark is necessary. The uncertainties for CHyM time series would suggest a good ensemble variability obtained by forcing CHyM with the WRF ensemble members, albeit the reduced variability found for the WRF regional ensemble time series (see Sect. 3.2). A possible explanation may be found in the pointwise comparison between the models and the observations. Generally, the pointwise comparison of the precipitation between the forecast and observation is penalizing for the NWP forecast. In this case, the WRF regional ensemble is at 9 km making the pointwise comparison even more penalizing. This is not the case for the time series of a catchment which accounts for all the upstream flow making the comparison not pointwise.

## 6 Conclusions

On 15 November 2017 a severe hydrological event hit the Abruzzo region, causing damages with a high social and economic impact on human activities. This event is used to investigate the reliability of a meteorological–hydrological ensemble chain. An operational and portable meteorological–hydrological forecast system is implemented and tested at CETEMPS over the Abruzzo region basins. The results of both the meteorological regional ensemble and the coupled meteorological–hydrological ensemble chain are discussed and compared with the observations and radar SRT. The results of a high-resolution deterministic simulation are used as a benchmark. The meteorological ensemble correctly reproduces the signal of the event by catching the area of the maximum precipitation a few days before the event. An overestimation of the maximum of the precipitation is found for the southern side of the Abruzzo region. The statistical evaluation using the ROC based on rain gauges and SRT supports this conclusion by showing a large AUC for the WRF regional ensemble and a steep increase of the POD. The meteorological–hydrological ensemble chain results are discussed in terms of the hydrological BDD stress index able to identify catchment segments that are most likely to be stressed by weather extremes. The evaluation of the BDD index, as a user-oriented instrument to assess the flood risk over the Abruzzo region drainage network, is carried out by comparing the occurrence of the index threshold exceedance in CHyM-OBS and the corresponding water level thresholds exceedance at station level. The results show a good agreement. In this regard, it should be taken into account that the BDD index obtained from CHyM-OBS is used as reference product for the BDD index itself and the

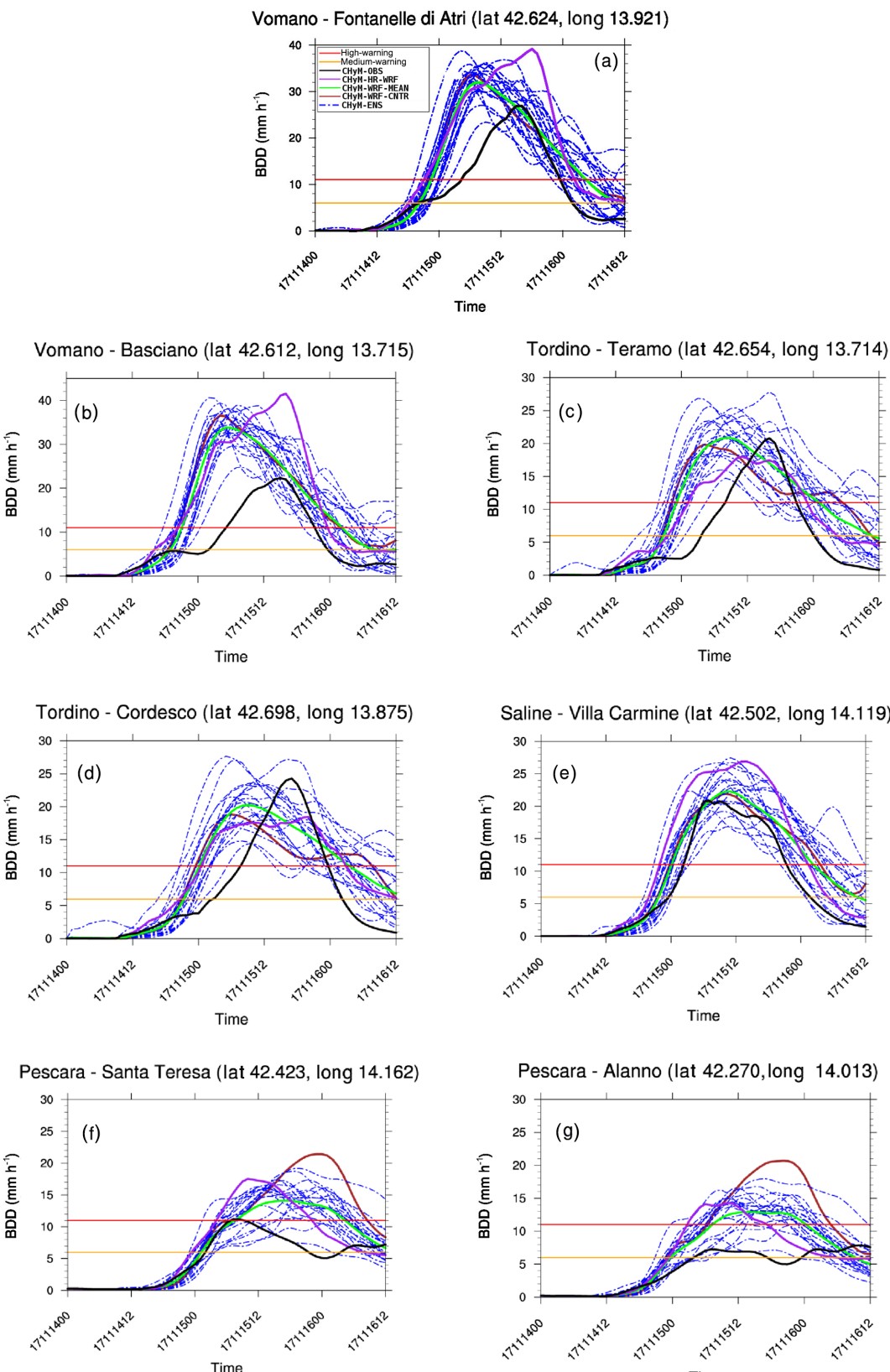

**Figure 13.** BDD time series (time given in the format of day, month, two-digit year and hour) for catchments of Vomano, Tordino, Saline and Pescara at stations **(a)** Fontanelle di Atri (Vomano), **(b)** Basciano (Vomano), **(c)** Teramo (Tordino), **(d)** Cordesco (Tordino), **(e)** Villa Carmine (Saline), **(f)** Santa Teresa (Pescara) and **(g)** Alanno (Pescara).

meteorological–hydrological ensemble chain because of a lack of discharge estimations and of updated rating curves. Moreover, the BDD thresholds are extendible to each grid point of the drainage network and are used to produce a hydrological-stress map over the whole spatial domain. The BDD maps produced by several CHyM simulations initialized using different WRF outputs are also compared with the hydrogeological-criticality bulletin released during the event, in order to emphasize and confirm the spatial coherence between hydrological control simulation and the detection of actual flooded areas. A very good performance of the BDD index (for both maps and time series) is found using CHyM-WRF-MEAN and CHyM-HR-WRF. Besides the BDD index, a BDD probability index and the associated spread are built using the 21 members of CHyM-ENS. The index allows for estimating the probability of a flooding event, which is not possible to estimate by both the deterministic forecast and the CHyM forecast forced using the ensemble mean. The comparison of the $BDD_{prob}$ map with the BDD map produced by CHyM-OBS points out a good reliability of this index for this event by both correctly identifying flooded river segments and producing a small spread in these areas.

Hence we can summarize the major findings as follows:

– The pseudo-hydrological ensemble forecast, i.e. the hydrological model forced by the mean precipitation produced by the WRF regional ensemble, reproduces the alert map issued by the Civil Protection Department during the event well, conferring reliability to this tool.

– The CHyM ensemble composed of 21 members and initialized using the 21 members of the WRF regional ensemble allows for computing the probabilistic BDD maps, producing information on the reliability of the event. Moreover, the $BDD_{prob}$ index map agrees well with the alert map issued by the Civil Protection Department during the event, slightly reducing the overestimation produced by the pseudo-ensemble, especially on the southern side of the Abruzzo region.

– The CHyM ensemble compares well with the CHyM simulation forced using the HR deterministic forecast.

Therefore, the results indicate advantages in using a meteorological–hydrological ensemble prediction chain, especially in terms of decision support system (DSS) efficiency. Specifically, the possibility of producing a probabilistic index informing about potentially flood prone areas earlier is a relevant tool for early-warning decision makers. Moreover, its low computational cost if compared with an HR deterministic modelling chain makes it affordable even to the small centres.

It has to be pointed out that the alert map was issued on the morning of 15 November (i.e. the same day of the event) by the Civil Protection Department (DPC) by initially using CHyM-HR-WRF, and then it was updated using the observations. Hence, the availability of a forecast well in advance

(i.e. at least the day before the event) would allow the DPC to issue an alert map the day before. This can be achieved by using an ensemble forecast which, though at lower resolution, produces both a forecast for a longer lead time than the deterministic and information on the probability of the forecasted event.

Finally, an attempt is made to estimate the uncertainties in the precipitation forecast and how their errors propagate in the hydrological model if an ensemble approach is adopted. To this purpose a comparison is made between the BDD index time series extracted at station level, computed using CHyM-OBS and CHyM-WRF-ENS precipitation and temperature, and using the discharge field average by CHyM-ENS. The results do not show differences, but the CHyM ensemble spread reproduces a distribution different than the WRF regional ensemble one. Indeed, a larger spread than for the WRF regional ensemble is found for most of the stations either for the maximum or the minimum of the precipitation. Hence we suppose that the weather prediction uncertainty propagates into the hydrological model. Therefore, to further investigate the propagation of uncertainty into the hydrological model, the same probabilistic approach used for the meteorological model should be applied to the hydrological one. In a forthcoming work, a sensitivity study to the hydrological model uncertainty that can be obtained by perturbing, for example, the geometry of the system or the model factors will be performed. Whereas, with the aim of improving the spread of the WRF ensemble, a sensitivity study on the impact of adding a few members of a multiphysics ensemble to this meteorological ensemble will evaluated.

*Data availability.* Underlined simulated and radar datasets are available upon request to the lead author. Data concerning the rain gauges datasets have been collected from the DEWETRA (Civil Protection Department) data portal (http://dewetra.cimafoundation.org/dewetra/, last access: 3 September 2019). The platform is accessible upon request to the Civil Protection Department. CE1

*Supplement.* The supplement related to this article is available online at: https://doi.org/10.5194/hess-24-1-2020-supplement.

*Author contributions.* RF, LS and VM conceptualized the project. RF, LS, VM, BT and AL created the methodology. RF, LS, VM, VC, AL, and BT performed the formal analysis. RF, GR, LS, VM, IM and BT took part in the investigation. RF, GR, BT and MV procured the resources. RF, LS, BT, and VC prepared the original draft of the paper. RF, GR, MV, BT, AL, LS, IM and VM reviewed and edited the paper. VM, LS, AL, and BT visualized the data. RF, GR, and MV supervised the project.

*Competing interests.* The authors declare that they have no conflict of interest.

*Acknowledgements.* The authors acknowledge the National Center for Atmospheric Research (NCAR) and the Center of Excellence in Telesensing of Environment and Model Prediction of Severe events (CETEMPS) for financial and computing resources, NCAR for the WRF-ARW source code, and the Italian Civil Protection Department for rain gauge data.

*Financial support.* This research has been supported by the Center of Excellence in Telesensing of Environment and Model Prediction of Severe events (CETEMPS).

*Review statement.* This paper was edited by Laurent Pfister and reviewed by two anonymous referees.

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

**Remarks from the language copy-editor**

CE1     Please note the slight changes.

**Remarks from the typesetter**

TS1     According to our standards, changes like this must first be approved by the editor, as data have already been reviewed, discussed and approved. Please provide a detailed explanation for those changes that can be forwarded to the editor. Please note that this entire process will be available online after publication. Upon approval, we will make the appropriate changes. Thank you for your understanding.