# Peer review of "A meteo-hydro regional ensemble forecast for early warning system over small Apennine catchments on Central Italy"

_Hydrology and Earth System Sciences, 2019_

## Referee Comment (RC1) · Anonymous Referee #1 · 26 Jun 2019

This manuscript presents a regional-scale flood forecasting system based on the combination of ensemble meteorological predictions with hydrological modeling. The hydro-meteorological chain is based on the dynamical downscaling of short-term (i.e., 3 days) GFS predictions using WRF-ARW and offline CHyM hydrological model simulations. The forecasting system is tested for an extreme flood event that hit central part of Italy on November 2017. Flood warnings are issued using an index based on the ratio between the flow discharge and the hydraulic radius.

Establishing and testing an advanced flood warning system is a subject of interest for the broad hydro-meteorological community and certainly suitable for HESS readership.

[Figure]

Furthermore, water agencies operationally dealing with flood predictions should greatly benefit from well-designed and technically advanced studies aimed at investigating the skills and/or deficiencies of hydro-meteorological prediction chains. Unfortunately, this works does not offer new and useful insights, lacking novelty and a rigorous evaluation approach that could eventually turn into valuable information for both the scientific and operational hydro-meteorological communities. In light of this overall assessment, I would consider this manuscript not suitable for publication in HESS. I highlight below a series of more specific comments.

1. The manuscript fails to identify an outstanding research question associated to the development and testing of probabilistic flood forecasting. In my opinion, the novelty of the work cannot rely on the simple combination of downscaled probabilistic meteorological forecasts with hydrological simulations. Moreover, after reading the manuscript it is not clear what's the message the authors want to convey when they generically discuss about the pros and cons of deterministic vs probabilistic forecasting approach. Nowadays the combined use of both is an established practice implemented by many hydro-meteorological centers. That is, the "complementarity" should be substantiated with ad-hoc results and not just advocated. In addition, saying that the probabilistic approach allows for longer forecast periods is absolutely misleading. Finally, in order to put this work in the right perspective I would have referred to the recent and innovative efforts behind the development of WRF-Hydro aiming at fully integrated hydro-meteorological forecasts.

2. The manuscript lacks a quantitative approach in the analysis/interpretation of the modeling results. A robust verification framework should be part of a reliable early warning system. Several common verification scores should have been implemented in order to assess the performance of the modeling results; mainly precipitation and streamflow. Here I would have also paid special attention on the quantification of the spatial agreement between simulated and observed precipitation (using radar data), which is key for short-term distributed flood forecasting. The reader is left with unverifiable statements (e.g., "very good response", "good agreement", "suggesting a more accurate forecast", etc.) that do not make things clearer.

3. The content of the manuscript is not well organized (i.e., section order). In general you should present first data and methods (i.e., numerical models), define the skill metrics and/or indices, and at the end you interpret/discuss the results. The language should be improved. This is not just a matter of typos, grammar mistakes, and unclear sentences. For instance, "reliability" and "uncertainty" do not have the same meaning in probabilistic forecasting. I am not sure authors are using the term "spatial coherence" in the right way. Finally, many figures (e.g., Fig. 3, Fig. 7, Fig. 11) and Table 1 are really not necessary. Please note also that the geographical location of the two Italian regions (i.e., Umbria and Abruzzo) is not shown in any map. The same apply for the hydrometric stations.

4. I have some remarks on the model setup and configuration: - The use of 1° GFS forecast is not fully justified in my opinion because other deterministic and probabilistic products (e.g., ECMWF) are available at higher resolution. - Domain definition, grid resolution, and physical parameterizations have a large influence on model results. Did the authors made preliminary tests to check their impacts on the selected events? This is key for a sound modeling strategy, especially from an operational perspective. - The definition of the benchmark configuration is not clearly discussed. If I look at Fig. 1 of Pichelli et al., 2017, the 1km domain (D3) does not fully cover the study area of this work, am I wrong? In the same paper it is mentioned that the operational setup of WRF-CETEMPS is different from the one shown in Fig.1. Further, it seems that the high-resolution setup uses GFS later boundary forcing with different resolutions (i.e., 0.25° instead of 1° resolution) and different physical model parameterizations. Finally, it seems not completely justified to directly nest the 9km WRF into the 1° GFS forecast. I would have expected an intermediate step to reduce lateral boundary effects. In general, these aspects of the work are not clearly explained.

5. I have several remarks concerning the adopted discharge index and the discussion

of the related results: - Authors consider the definition of the BDD index necessary due to the lack of discharge measurements. This contradicts the definition (Eq. 2) of the index itself, which is based on discharge values! - What's the equation used to calculate the hydraulic radius as a function of the drainage area? - The comparison between Fig. 10 and Fig. 9 is not intuitive. - I do not fully agree with the interpretation of Fig. 10. I see a good agreement in the timing even for those stations heavily impacted from hydropower production (i.e., Vomano and Todino). I also think that the mismatch for Pescara River could be due to some error in the observed atmospheric forcing at the local stations. That's why a more careful evaluation of the atmospheric forcing would have provided more useful insights.

6. It is really difficult to follow the discussion around Fig. 12. For instance, authors interpret the results saying that the mismatch between "observed" and "simulated" BDD index is due to precipitation occurring only on a very small area and not capture by the model. What do you mean with "small"? The high-resolution simulations are at 1km! I suspect that you can get the same issue if you go down to 100m resolution. Again, if you do not carefully evaluate the atmospheric simulations it is difficult to provide convincing interpretation of the BDD index. Finally, I would also remark that authors talk about "overestimation" and "underestimation" of the BDD index using as a reference the model results driven with observed (interpolated?) precipitation. I am fine with this as long as you cross-validate local precipitation measurements with other sources of information, e.g., spatially distributed information obtained from radar retrievals.

7. I would expect the same kind of curve when I look at the black ("observed") lines in Fig. 14 and the red ones in Fig. 10, am I wrong somewhere? One of the main conclusions is that the uncertainty in the BDD index is underestimated if you do not perturb the parameters of the hydrological model. This is intuitive and this is the reason why you should take both ("atmospheric" and "hydrologic") into account. In my opinion this opportunity was missed in this work.

---

## Referee Comment (RC2) · Anonymous Referee #2 · 8 Jul 2019

The manuscript aims at assessing the relevance of applying ensemble simulations to meteo- and hydro-logical modelling to improve the forecast of flood events. The work is quite interesting, even if the statistical analysis of the ensemble and of the results is rather weak and some conclusions are not fully supported by the research outcomes.

I think that the paper can be improved with an accurate major revision.

Please also note the supplement to this comment:
https://www.hydrol-earth-syst-sci-discuss.net/hess-2019-223/hess-2019-223-RC2-supplement.pdf

[Figure]

**Supplement:**

Review of the manuscript HESS-2019-223
"Regional ensemble forecast for early warning system over small
Apennine catchments on Central Italy"
by Ferretti R, Lombardi A, Tomassetti B, Sangelantoni T, Colaiuda
V, Mazzarella V, Maiello I, Verdecchia M, Redaelli G

08 July 2019

**1   General comments**

The manuscript aims at assessing the relevance of applying ensemble simulations to meteo- and hydro-logical modelling to improve the forecast of flood events. The work is quite interesting, even if the statistical analysis of the ensemble and of the results is rather weak and some conclusions are not fully supported by the research outcomes.

I think that the paper can be improved with an accurate major revision, in order to fix the scientific flaws listed in the specific comments below.

**2   Specific comments**

1. The description of the ensemble is quite confusing to me. Throughout the paper, ensembles of 20 members and of 21 members are often mentioned (at page 2, lines 14 to 32; page 3, lines 30 to 32; page 6, lines 11 to 13; page 17, lines 9 & 10; etc.), but at the end I was not sure to have understood the difference between the two ensembles.

2. The statistical description of the ensemble and of the results is very weak. Only few, very basic statistics are considered and they are defined in a rather cumbersome way (see technical comments # 31 & 32).

3. Section 3.2 "Ensemble precipitation time series" should be discussed in a more accurate way.

   - I am sorry, but I think that the agreement between forecast and observations is not so exciting. In fact, I do not agree with the sentence "The meteorological ensemble well reproduces the event in terms of heavy precipitation area identification, as well as its onset and length" in the conclusions (Page 19, lines 21 & 22). A qualitative assessment would be more objective and the reader could decide whether the agreement is satisfactory or not.
   - The variability among ensemble members appears much smaller than the difference between ensemble mean and observations (Figure 6). This fact is not sufficiently considered, quantified and discussed in the text.

4. Page 20, lines 7 to 15. These conclusions should be reinforced. After a first, possibly fast, reading, I asked myself: "what is the relevance of the proposed method, if it merely confirms the results of the methods already in use by the civil protection agency?". Instead, the added value of the proposed method should be better discussed.

5. A few grammar and language errors should be fixed. They are listed in the technical comments.

**3   Technical comments**

1. Page 1, line 2. Correct "resolution". Rephrase "newly developed" in order to be more explicit.

2. Page 1, line 5. Modify "ensemble system" in order to clarify it.

3. Page 1, line 8. Substitute or specify "period".

4. Page 1, lines 11 & 12. Rephrase and improve "and of the uncertainty of this flood".

5. Page 1, line 15. Substitute "one of" with "among". Correct "estimated".

6. Page 1, line 20. Rephrase "large gradients", possibly by adding "of meteorological quantities". Rephrase "the cooler atmosphere and the warmer sea": which is the comparison term? cooler and warmer than what?

7. Page 1, line 23. Rephrase "Recent decades".

8. Page 2, line 2; page 25, line 24. Check the publication date of Van den Besselaar et al. (2011).

9. page 2, lines 3 & 4. Rephrase "a warmer atmosphere and a greater amount of water vapor": which is the comparison term?

10. Page 2, line 4. Check spelling of "Willet".

11. Page 2, line 6. Substitute "to" with "on".

12. Page 2, line 9. Is "occur" the proper word? May be, "are expected"?

13. Page 2, line 10. EU Flood Directive (2007) is referenced wth a different "author name" in the reference list. Add "," after "2007)". Provide references for "Recent studies".

14. Page 2, line 12. Is "lead time" the right expression?

15. Page 2, line 15. "Assuming an appropriate hydrological model formulation": this is not a weak assumption, this should be discussed more accurately.

16. Page 2, line 16. Correct "scales".

17. Page 2, line 23. Substitute "on" with "in".

18. Page 2, line 25; page 8, lines 13 & 17; page 19, line 9. Add "with" before "respect".

19. Page 2, line 26. "their added values belong" or "their added value belongs".

20. Page 3, line 10. Erase "ing" from "fostering".

21. Page 3, line 11. Substitute "placed in", possibly with "associated to".

22. Page 3, line 17. The GFS acronym has been introduced without explanation.

23. Page 3, lines 23 to 25. Rephrase the sentence "The results... extreme events".

24. Page 3, line 26. Substitute "as it has been discussed by", possibly with "on the basis of the results of". This topic should be discussed in more detail.

25. Page 4, line 4. Add "," after "trough".

26. Page 4, lines 9 & 10. Rephrase "The thermal advection... at the upper ones": thermal advection is a physical process and I am afraid that is not correct to associate this expression to adjectives like "warm" and "cold".

27. Page 4, line 11. Please replace "13 November 2017" with a more precise definition of the timing of the two phases.

28. Page 4, line 12. Please rephrase "not shown". Do you mean, not represented in the figures or in some of them? Or not discussed in the paper?

29. Page 4, line 18. Substitute "because of", possibly with "as evidenced by".

30. Page 4, line 27. Substitute "Similarly to", possibly with "Following" or "In accordance with". Erase "," after "study".

31. Page 8, lines 13 to 15. In the statistical literature, this is simply called the "ensemble standard deviation".

32. Page 8, lines 19 & 18. In the statistical literature, this is simply known as the "coefficient of variation".

33. Page 8, line 21. Add "it" before "has been".

34. Page 8, line 28. Substitute "this latter"; it is not clear.

35. Page 10, line 25. Check spelling of "Lighthill".

36. Page 10, line 32. The word "based" is repeated twice at short distance.

37. Page 13, line 3. Erase "," after "comparison". Erase "even".

38. Page 13, line 4. Rephrase "hourly data in m".

39. Page 20, line 11. Correct "initialized".

40. Page 20, lines 19 & 20. Rephrase "as a complementary tools by using both".

41. Page 23, line 20. Check "Coauthors".

42. Page 24, lines 11 to 14. These references are cited with a different, abbreviated author's name in the text.

43. Figure 1. Add contour spacing for the two represented quantities in the figure captions. Expand the acronym "mslp". Black lines are hardly visible; I think that they would be more visible if drawn in yellow.

44. Figure 2. I am afraid that it will be very difficult to read the legend in the printed version of the paper. In the caption, add a space in "24 hours".

45. Figure 6. I think that a better choice of the colours could help to examine the plots. In particular, inverting the colours for the Ensemble members and the Ensemble mean could help to visualize the mean more easily.

46. Table 1. The first column can be erased because it is not informative: it has the same value for all the simulation types. I think that it could be useful to add a column to assign a code to each simulation type.

---

## Author Comment (AC1) · 19 Jul 2019

**Answer to reviewer # 1**

**'Unfortunately, this works does not offer new and useful insights, lacking novelty and a rigorous evaluation approach that could eventually turn into valuable information for both the scientific and operational hydro-meteorological communities.'**

We believe that the reviewer does not sufficiently justify his/her statement:
- **lacking of novelty**
  *To our knowledge there are not yet coupled operational system using regional meteorological ensemble and hydro ensemble in Italy.*
  *To our knowledge there are approximately only a few Meteorological Centers producing Hydrological operational and pre-operational forecast using ensemble prediction weather forecast in Europe (Cloke and Pappenberger, 2009)*
- **rigorous evaluation approach**
  *Actually, the aim of the paper is not the statistical evaluation of an operational system but its application to a case study. We believe that a statistical evaluation based on only one case study is not scientifically consistent. However, we agree in using a statistical approach to objectively evaluate the response of the ensemble as we have already done for other cases study (Serafin and Ferretti, 2007, Maiello et al., 2014,. Maiello et al., 2017)*

1. **'In my opinion, the novelty of the work cannot rely on the simple combination of downscaled probabilistic meteorological forecasts with hydrological simulations. Moreover, after reading the manuscript it is not clear what's the message the authors want to convey when they generically discuss about the pros and cons of deterministic vs probabilistic forecasting approach. Nowadays the combined use of both is an established practice implemented by many hydro-meteorological centers. That is, the "complementarity" should be substantiated with ad-hoc results and not just advocated. In addition, saying that the probabilistic approach allows for longer forecast periods is absolutely misleading. Finally, in order to put this work in the right perspective I would have referred to the recent and innovative efforts behind the development of WRF-Hydro aiming at fully integrated hydro- meteorological forecasts.'**

   *As in the previous statement, we think that the reviewer does not sufficiently justify his/her sentences:*
   - **the novelty of the work cannot rely on the simple combination of downscaled probabilistic meteorological forecasts with hydrological simulations**
     *We probably missed to clearly specify that this is an off-line coupling of the regional ensemble weather forecast and the hydrological ensemble forecast for the Italian regions. To our knowledge there are not Italian weather forecast centers performing this kind of forecast. Please, let us know which are the hydro-meteorological centers using ensemble weather forecast and ensemble hydrological forecast so that we can refer to them.*
   - **Moreover, after reading the manuscript it is not clear what's the message the authors want to convey when they generically discuss about the pros and cons of deterministic vs probabilistic forecasting approach. Nowadays the combined use of both is an established practice implemented by many hydro-meteorological centers.**
     *We will clarify this aspect in our conclusions.*
   - **That is, the "complementarity" should be substantiated with ad-hoc results and not just advocated.**
     *We will add an objective evaluation of the ensemble forecast to support our conclusions.*
   - **In addition, saying that the probabilistic approach allows for longer forecast periods is absolutely misleading.**
     *We do not agree with sentence. Most of the international weather forecast centers (ECMWF, NCEP etc.) perform ensemble forecast for longer periods than using the high resolution deterministic forecast. Please justify this remark.*
   - **Finally, in order to put this work in the right perspective I would have referred to the recent and innovative efforts behind the development of WRF-Hydro aiming at fully integrated hydro-meteorological forecasts.**
     *We do agree that the comparison with a well-established and well known (for a long time) hydrological model (WRF-Hydro) fully coupled with the weather forecast is an interesting point to investigate but it is another study. Moreover, we do believe that a different choice regarding the hydrological model*

*(well referenced too: Tomassetti et al., 2005, Coppola et a., 2007, Verdecchia et al., 2008) does not represent a weakness of the present study.*

2. **'The manuscript lacks a quantitative approach in the analysis/interpretation of the modeling results. Several common verification scores should have been implemented in order to assess the performance of the modeling results; mainly precipitation and streamflow. Here I would have also paid special attention on the quantification of the spatial agreement between simulated and observed precipitation (using radar data), which is key for short-term distributed flood forecasting'**
   - **Several common verification scores should have been implemented in order to assess the performance of the modeling results;**
     *As already stated at the beginning, we did not perform specific statistical evaluation because the aim of the paper is not the statistical evaluation of an operational system but the application of the system to a case study. We believe that performing a statistical evaluation for a case study only is not scientifically supported. However, a quantitative statistical approach to objectively evaluate the response of the ensemble will be added in the revised version of the manuscript.*
   - **Here I would have also paid special attention on the quantification of the spatial agreement between simulated and observed precipitation (using radar data)**
     *We will do this comparison. Just to point out, the radar precipitation is a derived product that may have its own error as well. That is why we used rain gauges. However, we will accordingly add the radar dataset and both the products will be considered in the revised manuscript version.*

3. **'The content of the manuscript is not well organized (i.e., section order). In general, you should present first data and methods (i.e., numerical models), define the skill metrics and/or indices, and at the end you interpret/discuss the results. The language should be improved. This is not just a matter of typos, grammar mistakes, and unclear sentences. Finally, many figures (e.g., Fig. 3, Fig. 7, Fig. 11) and Table 1 are really not necessary. Please note also that the geographical location of the two Italian regions (i.e., Umbria and Abruzzo) is not shown in any map. The same apply for the hydrometric stations. '**

   **- The content of the manuscript is not well organized (i.e., section order).**
   *The organization of the paper is basically what the Reviewer#1 is suggesting. We first presented the case study and the data. Then, it follows the methods used i.e. the models.*
   *We decided to separate the presentation of the models and their results. WRF and its results presented first and then it follows CHyM with related results.*
   *Therefore, the paper organization is the following:*
   *1. Introduction*
   *2. Case study, that is the data used for this study, as you are suggesting*
   *3. WRF Numerical model, that is the method used for the weather forecast*
      *-Ensemble weather forecast results*
   *4. CHyM Hydrological model, that is the method used for the hydrological forecast*
      *- Method used to evaluate the hydrological results: BDD index*
   *5. Hydrological model results using different forcing: ensemble mean, all members and deterministic*
   *6. Conclusions*

   - **The language should be improved.**
     *We will carefully review the whole paper paying particular attention to the grammar and the language.*
     **- Finally, many figures (e.g., Fig. 3, Fig. 7, Fig. 11) and Table 1 are really not necessary. Please note also that the geographical location of the two Italian regions (i.e., Umbria and Abruzzo) is not shown in any map. The same apply for the hydrometric stations.'**
     *We agree with you, Figs. 3, 7 and 11 will be removed from the paper. We believe that table 1 helps understanding the performed experiments. We will add the location of the Italian regions Umbria and Abruzzo.*

4. **'I have some remarks on the model setup and configuration: - The use of 1° GFS forecast is not fully justified in my opinion because other deterministic and probabilistic products (e.g., ECMWF) are available at higher resolution. - Domain definition, grid resolution, and physical parameterizations have a large influence on model results. Did the authors made preliminary tests to check their impacts on the selected events? This is key for a sound modeling strategy, especially from an operational perspective. - The definition of the benchmark configuration is not clearly discussed. If I look at Fig. 1 of Pichelli et al., 2017, the 1km domain (D3) does not fully cover the study area of this work, am I wrong? In the same paper it is mentioned that the operational setup of WRF-CETEMPS is different from the one shown in Fig.1. Further, it seems that the high-resolution setup uses GFS later boundary forcing with different resolutions (i.e., 0.25° instead of 1° resolution) and different physical model parameterizations. Finally, it seems not completely justified to directly nest the 9km WRF into the 1° GFS forecast. I would have expected an intermediate step to reduce lateral boundary effects. In general, these aspects of the work are not clearly explained.**

- **'I have some remarks on the model setup and configuration: - The use of 1° GFS forecast is not fully justified in my opinion because other deterministic and probabilistic products (e.g., ECMWF) are available at higher resolution. - Domain definition, grid resolution, and physical parameterizations have a large influence on model results. Did the authors made preliminary tests to check their impacts on the selected events?**

  *Based on our long experience in using several numerical models (MM4, MM5, WRF, Harmonie), we run operational weather forecast since 1998 (Paolucci et al., 1999) and several paper published on this topic, we defined the model set up and performed several experiments using different ICs (ECMWF and GFS) and different parameterizations. The best results we end up with is the one presented in this paper.*

  *Unfortunately, the NCEP archive allows for retrieving only 1° forecast and analysis, this why we used the 1° GFS. For what concerns ECMWF we performed several tests using the 50 members of ECMWF ensemble at 0.125 but the results obtained were not satisfactory.*

- **'The definition of the benchmark configuration is not clearly discussed. If I look at Fig. 1 of Pichelli et al., 2017, the 1km domain (D3) does not fully cover the study area of this work, am I wrong?**

  *The Pichelli's work is on the Pò Valley that is on a Valley delimited by the Mountains (Alps and Apennine) that is complex orography region as Abruzzo region is. Therefore, based on the Pichelli's work we set up the deterministic operational forecast at 1km ([http://magritte.aquila.infn.it/meteo/ecmwrf-2way/](http://magritte.aquila.infn.it/meteo/ecmwrf-2way/)) over Abruzzo region. We used this operational deterministic configuration to run an 'ad hoc' deterministic forecast using 0.25 GFS, ICs and BCs, for this event and we used it as benchmark.*

- **In the same paper it is mentioned that the operational setup of WRF-CETEMPS is different from the one shown in Fig.1.**

  *Please see the previous answer.*

- **Further, it seems that the high-resolution setup uses GFS later boundary forcing with different resolutions (i.e., 0.25° instead of 1° resolution) and different physical model parameterizations.'**

  *Yes, we used the best deterministic forecast produced by GFS at 0.25° and the best available GFS ensemble forecast, that is at 1°. Since we used the deterministic high-resolution forecast as benchmark we decided to do not downgrade it. The use of different parameterizations, as you just stated, is driven by the different resolution. We have to use a cumulus convection parameterization at 9km which is not necessary at 1km because convection is explicitly resolved at this resolution.*

- **'I would have expected an intermediate step to reduce lateral boundary effects. In general, these aspects of the work are not clearly explained.'**

  *We agree with you. Generally, it is better to use an intermediate domain, if going down from 1° to 9km, for providing BCs to the nested domain. We performed several simulations but there was not any improvement in the results with respect to the direct nesting into the GFS ICs. Therefore, based on these results we decided to perform the ensemble forecast directly nesting the 9km to the 1° GFS ICs. We will add an explanation about this in the reviewed version of the paper.*

5. 'I have several remarks concerning the adopted discharge index and the discussion of the related results: - Authors consider the definition of the BDD index necessary due to the lack of discharge measurements. This contradicts the definition (Eq. 2) of the index itself, which is based on discharge values! - What's the equation used to calculate the hydraulic radius as a function of the drainage area? The comparison between Fig. 10 and Fig. 9 is not intuitive. - I do not fully agree with the interpretation of Fig. 10. I see a good agreement in the timing even for those stations heavily impacted from hydropower production (i.e., Vomano and Todino). I also think that the mismatch for Pescara River could be due to some error in the observed atmospheric forcing at the local stations. That's why a more careful evaluation of the atmospheric forcing would have provided more useful insights.**

- **Authors consider the definition of the BDD index necessary due to the lack of discharge measurements. This contradicts the definition (Eq. 2) of the index itself, which is based on discharge values! - What's the equation used to calculate the hydraulic radius as a function of the drainage area?**
  *The observation is very appropriate and we thank the referee for this; we tried to summarize in one sentence two different problems, leading to a lot of confusion. The first problem deals with the difficulties to calibrate the discharge predicted by any hydrological model with observed data. Discharge observations in continuous time series are often missing, especially for small basins.*
  *A different problem is to use the predicted discharge for flood alert mapping, as it is not straightforward to establish a threshold level above which a critical event is to be expected; in addition such threshold level should be calculated for each grid point because it depends on the size of the river bed in the selected point. To overcome this second problem we tested different general definition of an alarm index and, after simulating different case studies occurring in different basins of different size, we find that a suitable definition could be the ratio between the maximum value of the predicted discharge within a given time interval and the square of hydraulic radius that is a "measure" of the river cross section for the selected point. The definition of BDD index has also a simple physical interpretation: it represents the average precipitation (more specifically the precipitation available for the runoff) drained by each grid element from the upstream basin.*
  *The BDD index is based on Eq. 2: in this equation, the used discharge value is not the measured value, but the discharge computed by the CHyM model, forced with observed raingauges data as input.*
  *As for many other models (for a general reference see Singh and Frevert, 2002) the hydraulic radius can be approximated as a linear function of drained area. In particular $R=\beta+\gamma D^{\delta}$ where $\beta$, $\gamma$ and $\delta$ are empirically established and the value of $\delta$ is very close to 1. If the area is measured in Km2, typical values taken from literature are $\beta=0.0015$ and $\gamma=0.05$, while (for a general reference see Singh and Frevert, 2002).*

- **The comparison between Fig. 10 and Fig. 9 is not intuitive. - I do not fully agree with the interpretation of Fig. 10. I see a good agreement in the timing even for those stations heavily impacted from hydropower production (i.e., Vomano and Todino).**
  *We will explain in details the two figures. For what concerns fig.9 the four red triangle-shaped, thin-bounded signs indicates the relevant hydrometers where the red hydrometric threshold has been exceeded; in particular, among the involved rivers, there are Vomano, Tordino, Saline and Pescara. In figure 10, the normalized water level and BDD time series along the aforementioned rivers, for different hydrometric station grid-points, are shown.*
  *Generally, hydroelectric power installations can heavily impact the flood dynamics along a river basin, but the key parameters to be considered are various , such as the relative importance of the drained areas, the water storage capacity and the position of the reservoirs within the basin. Nevertheless, the effect highly depends on the initial reservoir filling rate, which is unknown. If the reservoirs are already full before a flood, no (gated spillway) or limited (ungated spillway) flood routing is possible (Jordan et al.,2012). Unfortunately, in Abruzzo region we are not aware of how the hydroelectric systems are managed. In this particular case, the hydroelectric power plants of Provvidenza and Piaganini are located upstream (Figs. 1 and 2, below this section), respect to the areas involved in the event, where also precipitation maxima occurred. Probably, in this case, the effect of the hydroelectric power plants is*

*negligible and this sentence is confirmed by the good agreement in the timing shown by Fig.10.*

[Figure]

Figure 1: The map represents the accumulated precipitation from 0 UTC to 12 UTC on 15th Nov 2017, as measured by the raingauges network and spatialized over the region by using the Cellular Automata-based techniques. The area enclosed in the red line is the boundary of the Tordino basin. Blue triangles indicate the position or the Provvidenza and Piaganini dams.

[Figure]

Figure 2 is a zoom of the figure 1, where the Tordino drainage network is indicated by the blue     lines. Together with the dams, relevant hydrometric station are also indicated through red pinpoints.

- **I also think that the mismatch for Pescara River could be due to some error in the observed atmospheric forcing at the local stations. That's why a more careful evaluation of the atmospheric forcing would have provided more useful insights.**
  *Please clarify what you mean by 'error in the observed atmospheric forcing'. Are you referring to the data quality at Pescara station?*

6. **For instance, authors interpret the results saying that the mismatch between "observed" and "simulated" BDD index is due to precipitation occurring only on a very small area and not capture by the model. What do you mean with "small"? The high-resolution simulations are at 1km! I suspect that you can get the same issue if you go down to 100m resolution. Again, if you do not carefully evaluate the atmospheric simulations it is difficult to provide convincing interpretation of**

the BDD index. Finally, I would also remark that authors talk about "overestimation" and "underestimation" of the BDD index using as a reference the model results driven with observed (interpolated?) precipitation. I am fine with this as long as you cross-validate local precipitation measurements with other sources of information, e.g., spatially distributed information obtained from radar retrievals.

- **For instance, authors interpret the results saying that the mismatch between "observed" and "simulated" BDD index is due to precipitation occurring only on a very small area and not capture by the model. What do you mean with "small"?**
  *The Calvano river is a very small basin (35 km^2) and is located close to Vomano final segment, southward. The distance between the two rivers is about 2.5 km in the upper part of the Calvano's path and almost 6 km in the two mouths. Being so close, the rain spatial distribution plays a very important role: an error of even 1 km can significantly affect the forecast. Nevertheless, the Civil Protection early warning system is referred to "warning areas", rather than the single river segment or the single catchment area. For this reason, in a Decision Support System perspective, is important to assign the correct alarm state at warning area level, rather than meticulously focusing on the single catchment.*
- **Again, if you do not carefully evaluate the atmospheric simulations it is difficult to provide convincing interpretation of the BDD index.**
  *Please clarify this sentence. What do you mean by carefully evaluate atmospheric simulation? To objectively evaluate the weather forecast, as we already said, we will use skill statistical metrics, is this what you are suggesting?*
- **I am fine with this as long as you cross-validate local precipitation measurements with other sources of information, e.g., spatially distributed information obtained from radar retrievals.**
  *We will compare the results with the retrieved radar precipitation, but again being the radar precipitation a retrieved product it is affected by error as much as other observed parameters.*

7. **'I would expect the same kind of curve when I look at the black ("observed") lines in Fig. 14 and the red ones in Fig. 10, am I wrong somewhere? One of the main conclusions is that the uncertainty in the BDD index is underestimated if you do not perturb the parameters of the hydrological model. This is intuitive and this is the reason why you should take both ("atmospheric" and "hydrologic") into account. In my opinion this opportunity was missed in this work.**

   - **I would expect the same kind of curve when I look at the black ("observed") lines in Fig. 14 and the red ones in Fig. 10, am I wrong somewhere?**
     *The curves appear different because of the different temporal scale. Moreover, figure 10 shows the normalized index values, whereas fig.14 shows values in mm/h.*

   - **One of the main conclusions is that the uncertainty in the BDD index is underestimated if you do not perturb the parameters of the hydrological model. This is intuitive and this is the reason why you should take both ("atmospheric" and "hydrologic") into account. In my opinion this opportunity was missed in this work.**
     *Based on Cloke and Pappenberger (2009) this is not an intuitive conclusion. The lack of hydrological ensemble forecast does not allow to make such statement. If you are aware of different published conclusions please let us know.*

References

Jordan F.M., Boillat J.-L. and Schleiss A. J., Optimization of the flood protection effect of a hydropower multi-reservoir system, Intl. J. River Basin Management Vol. 10, No. 1, pp. 65 – 72, 2012

Maiello I., R. Ferretti, S. Gentile, M. Montopoli, E. Picciotti, F. S. Marzano, and C. Faccani: Impact of radar data assimilation for the simulation of a heavy rainfall case in central Italy using WRF–3DVAR. Atmos. Meas. Tech., 7, 2919–2935, 2014 www.atmos-meas-tech.net/7/2919/2014/ doi:10.5194/amt-7-2919-2014

Maiello I., S. Gentile, R. Ferretti, L. Baldini, N. Roberto, E. Picciotti, P. P. Alberoni, and F. S. Marzano: Impact of multiple radar reflectivity data assimilation on the numerical simulation of a flash flood event during

the HyMeX campaign. Hydrol. Earth Syst. Sci., 21, 5459–5476, 2017 https://doi.org/10.5194/hess-21-5459-2017

Searfin S. and R. Ferretti,: Sensitivity of a Mesoscale Model to Microphysical Parameterizations in the MAP SOP Events IOP2b and IOP8. JAMC, 46, 1438-1454, 2007. DOI: 10.1175/JAM2545.1

Singh VP, Frevert DK (2002) Mathematical models of smallwatershed hydrology and application. Water ResourcePublications, LLC, Highlands Ranch, Colorado, USA

---

## Author Comment (AC2) · 19 Jul 2019

**Answer to reviewer #2**

1. **The description of the ensemble is quite confusing to me. Throughout the paper, ensembles of 20 members and of 21 members are often mentioned (at page 2, lines 14 to 32; page 3, lines 30 to 32; page 6, lines 11 to 13; page 17, lines 9 & 10; etc.), but at the end I was not sure to have understood the difference between the two ensembles.**

   *Sorry for not making a clear statement on the ensembles. We performed an ensemble forecast using the GFS 20 members + the control; that is why we stated 21 members. We used the WRF 21 simulations to force the hydrological model producing 21 members for the CHyM ensemble. Therefore, the two ensembles are:*

   - *the WRF regional ensemble (forced by 20 GFS members + control)*
   - *the CHyM ensemble (forced by 21 WRF members).*

   *We hope this clarifies this point. We will accordingly clarify this part in the revised version of the manuscript.*

2. **The statistical description of the ensemble and of the results is very weak. Only few, very basic statistics are considered and they are defined in a rather cumbersome way (see technical comments # 31 & 32).**

   **31. Page 8, lines 13 to 15. In the statistical literature, this is simply called the "ensemble standard deviation".**

   **32. Page 8, lines 19 & 18. In the statistical literature, this is simply known as the "coefficient of variation".**

   *We will make more robust the description of the ensemble and the discussion of the results by adding some statistical evaluation to objectively compare the ensemble with the observation. Thanks for the corrections: following your suggestion we changed the* mean-related spread into ensemble standard deviation, *and accordingly we introduced the* coefficient of variation instead of the current definition.

3. **Section 3.2 "Ensemble precipitation time series" should be discussed in a more accurate way.**

   **· I am sorry, but I think that the agreement between forecast and observations is not so exciting. In fact, I do not agree with the sentence "The meteorological ensemble well reproduces the event in terms of heavy precipitation area identification, as well as its onset and length" in the conclusions (Page 19, lines 21 & 22). A qualitative assessment would be more objective and the reader could decide whether the agreement is satisfactory or not.**

   *As already mentioned a detailed and accurate statistical evaluation of the ensemble forecast will be performed to achieve a more objective conclusion than the one presented.*

   **· The variability among ensemble members appears much smaller than the difference between ensemble mean and observations (Figure 6). This fact is not sufficiently considered, quantified and discussed in the text.**

   *We agree with you, sorry for not having properly considered this point. We will further discuss it in the revised version of the paper by considering also difference between ensemble mean of the Hydrological forecast and observations which appears larger than the variability among the ensemble members at most stations (Figure 14).*

4. **Page 20, lines 7 to 15. These conclusions should be reinforced. After a first, possibly fast, reading, I asked myself: "what is the relevance of the proposed method, if it merely confirms the results of the methods already in use by the civil protection agency?". Instead, the added value of the proposed method should be better discussed.**

   *Again, sorry for not been clear over the mentioned statement. The alert map (Fig. 9), that we used as as 'ground truth'. This map was issued in the morning of Nov 15 (i.e. the same day of the event) by the Civil Protection Agency (CPA). The map is initially built on the deterministic forecast and then updated using observations. Therefore, the methods already in use by the civil protection agency is based on the deterministic models: the hydrological forecast forced by the meteorological forecast. This method is a very useful tool, but with short outlook. The availability of a forecast product well in advance (i.e. at least the day before of the event) would allow the CPA to issue an alert map the day before. This can be achieved by using an ensemble forecast which, though at lower resolution, produces both a forecast for a longer lead time than the deterministic, and information on the probability of the forecasted event. We will accordingly improve the discussion of this point in the conclusions section.*

**Technical Points.**
1. Ok done.
2. Ok
3. Ok
4. Ok, thanks
5. Ok done
6. Ok
7. Ok. Done
8. Ok we will rephrase
9. Ok, done
10. Ok, done
11. Ok, done
12. You are right, we substituted 'occur' with 'is expected'.
13. Ok we will add references.
14. We change lead time into outlook.
15. Ok we will do it
16. Ok, done
17. Ok, done
18. Ok, done
19. Ok, done
20. Ok, done
21. Ok, done
22. The GFS acronym has been introduced.
23. Ok, we will do it
24. Ok, we changed the sentence and we will add more details in the discussion
25. Ok, done
26. Ok, we changed the sentence
27. Ok, we will do it
28. We mean not presented in the figure and not discussed. We will change the sentence
29. Ok, done
30. Ok, done
31. Ok, done
32. Ok, done
33. Ok, done
34. Ok, we substitute this latter with the control simulation
35. Ok, done
36. Ok, we changed the verb
37. Ok, done
38. Ok, done
39. Ok, done
40. Ok, done
41. Ok, done
42. Ok, done
43. We will correct fig.1
44. We will correct fig. 2
45. We will correct fig. 6
46. We agree, we will assign a code to each simulation and we will correct the text.

---

## Author Response (AR1)

**Answer to reviewer # 1**

**'Unfortunately, this works does not offer new and useful insights, lacking novelty and a rigorous evaluation approach that could eventually turn into valuable information for both the scientific and operational hydro-meteorological communities.'**

- **lacking of novelty**
  *To our knowledge there are not yet operational system based on coupled regional meteorological ensemble and hydro ensemble in Italy.*
  *Further, to our knowledge, there are approximately only a few Meteorological Centers producing Hydrological operational and pre-operational forecasts using ensemble prediction weather forecast in Europe (please refer to Cloke and Pappenberger, 2009).*
- **rigorous evaluation approach**
  *Actually, the aim of the paper is not a statistical evaluation of the operational system, but its application to a case study. However, we accordingly added an objective evaluation of the response of the ensemble through a statistical analysis as already performed for other study (Serafin and Ferretti, 2007, Maiello et al., 2014,. Maiello et al., 2017). The statistical analysis relies on the Receiver Operating Characteristic (ROC) curves to evaluate the ensemble forecast precipitation using both rain gauges and radar retrieved precipitation.*

1. **'In my opinion, the novelty of the work cannot rely on the simple combination of downscaled probabilistic meteorological forecasts with hydrological simulations. Moreover, after reading the manuscript it is not clear what's the message the authors want to convey when they generically discuss about the pros and cons of deterministic vs probabilistic forecasting approach. Nowadays the combined use of both is an established practice implemented by many hydro-meteorological centers. That is, the "complementarity" should be substantiated with ad-hoc results and not just advocated. In addition, saying that the probabilistic approach allows for longer forecast periods is absolutely misleading. Finally, in order to put this work in the right perspective I would have referred to the recent and innovative efforts behind the development of WRF-Hydro aiming at fully integrated hydro- meteorological forecasts.'**

   - **the novelty of the work cannot rely on the simple combination of downscaled probabilistic meteorological forecasts with hydrological simulations**
     *We probably missed to clearly specify that this is an off-line coupling of the regional ensemble weather forecast and the hydrological ensemble forecast for the Italian regions. To our knowledge there are not Italian weather forecast centers performing this kind of forecast.*
   - **Moreover, after reading the manuscript it is not clear what's the message the authors want to convey when they generically discuss about the pros and cons of deterministic vs probabilistic forecasting approach. Nowadays the combined use of both is an established practice implemented by many hydro-meteorological centers.**
     *We have re-written the final statements, pag. 14-16*
   - **That is, the "complementarity" should be substantiated with ad-hoc results and not just advocated.**
     *We added an objective evaluation of the ensemble forecast to support our conclusion using ROC at several thresholds for both rain gauges and radar retrieved precipitation data, par 3.3, pag 8.*
   - **In addition, saying that the probabilistic approach allows for longer forecast periods is absolutely misleading.**
     *Most of the international weather forecast centers (ECMWF, NCEP etc.) perform ensemble forecast for longer periods than using the high resolution deterministic forecasts.*
   - **Finally, in order to put this work in the right perspective I would have referred to the recent and innovative efforts behind the development of WRF-Hydro aiming at fully integrated hydro-meteorological forecasts.**
     *We agree that the comparison with a well-established and well known hydrological model (WRF-Hydro) fully coupled with the weather forecast is an interesting tool to investigate flood events but it will be considered in a future research works. Moreover, from our point of view, we do believe that a different choice regarding the hydrological model (please refer to: Tomassetti et al., 2005, Coppola et a., 2007, Verdecchia et al., 2008) cannot represent a weakness of the present study.*

2. **'The manuscript lacks a quantitative approach in the analysis/interpretation of the modeling results. Several common verification scores should have been implemented in order to assess the performance of the modeling results; mainly precipitation and streamflow. Here I would have also paid special attention on the quantification of the spatial agreement between simulated and observed precipitation (using radar data), which is key for short-term distributed flood forecasting'**

    - **Several common verification scores should have been implemented in order to assess the performance of the modeling results;**

        *As already stated at the beginning, we did not perform specific statistical evaluation because the aim of the paper is not the statistical evaluation of an operational system but the application of the system to a case study. However, we accordingly performed a quantitative statistical analysis to objectively evaluate the response of the ensemble. Please refer to paragraph 3.3 pag 8 of the revised version of the manuscript.*

    - **Here I would have also paid special attention on the quantification of the spatial agreement between simulated and observed precipitation (using radar data)**

        *We performed the ROC statistical analysis using also radar data sets that cover the entire geographical domain considered. Please refer to the text in section 3.3 of revised manuscript.*

3. **'The content of the manuscript is not well organized (i.e., section order). In general, you should present first data and methods (i.e., numerical models), define the skill metrics and/or indices, and at the end you interpret/discuss the results. The language should be improved. This is not just a matter of typos, grammar mistakes, and unclear sentences. Finally, many figures (e.g., Fig. 3, Fig. 7, Fig. 11) and Table 1 are really not necessary. Please note also that the geographical location of the two Italian regions (i.e., Umbria and Abruzzo) is not shown in any map. The same apply for the hydrometric stations. '**

    **The content of the manuscript is not well organized (i.e., section order).**
    *The organization of the paper is the following:*

    1. *Introduction*
    2. *Case study description and data used*
    3. *WRF Numerical model, that is the method used for the weather forecast*
       *-Ensemble weather forecast results*
    4. *CHyM hydrological model, that is the method used for the hydrological forecast*
       - *Method used to evaluate the hydrological results: BDD index*
    5. *Hydrological model results using different atmospheric forcing: WRF ensemble mean, the entire set of WRF ensemble members and WRF deterministic high resolution simulation.*
    6. *Conclusions*
    *which is similar to your request.*

    **-The language should be improved.**
    *We carefully reviewed the whole paper paying particular attention to the grammar and the language.*

    **- Finally, many figures (e.g., Fig. 3, Fig. 7, Fig. 11) and Table 1 are really not necessary. Please note also that the geographical location of the two Italian regions (i.e., Umbria and Abruzzo) is not shown in any map. The same apply for the hydrometric stations.'**
    *We agree, Figs. 3, 7 and 11 will be removed from the paper. We believe that Table 1 could help the reader understanding the performed experiments. We accordingly added the location of the Italian regions Umbria and Abruzzo in Figure 3.*

4. **'I have some remarks on the model setup and configuration: - The use of 1° GFS forecast is not fully justified in my opinion because other deterministic and probabilistic products (e.g., ECMWF) are available at higher resolution. - Domain definition, grid resolution, and physical parameterizations have a large influence on model results. Did the authors made preliminary tests to check their impacts on the selected events? This is key for a sound modeling strategy, especially from an operational perspective. - The definition of the benchmark configuration is not clearly discussed. If I look at Fig. 1 of Pichelli et al., 2017, the 1km domain (D3) does not fully cover the study area of this work, am I wrong? In the same paper it is mentioned that the operational**

**setup of WRF-CETEMPS is different from the one shown in Fig.1. Further, it seems that the high-resolution setup uses GFS later boundary forcing with different resolutions (i.e., 0.25° instead of 1° resolution) and different physical model parameterizations. Finally, it seems not completely justified to directly nest the 9km WRF into the 1° GFS forecast. I would have expected an intermediate step to reduce lateral boundary effects. In general, these aspects of the work are not clearly explained.**

- **'I have some remarks on the model setup and configuration: - The use of 1° GFS forecast is not fully justified in my opinion because other deterministic and probabilistic products (e.g., ECMWF) are available at higher resolution. - Domain definition, grid resolution, and physical parameterizations have a large influence on model results. Did the authors made preliminary tests to check their impacts on the selected events?**
  *Based on our long experience in using several numerical models (MM4, MM5, WRF, Harmonie), we run operational weather forecast since 1998 (Paolucci et al., 1999) and several papers published on this topic, we defined the model set up and performed several experiments using different initial and boundary conditions (ICs Bcs) (ECMWF and GFS) and different parameterizations. The best results we end up with is the one presented in this paper. Unfortunately, the NCEP archive only allows for retrieving 1-deg.-resolution forecast and analysis. That is why we used the 1 deg. GFS ensemble as ICs and Bcs for the WRF ensemble. For what concerns ECMWF we performed several tests using the 50 members of ECMWF ensemble at 0.125 but the results obtained were not satisfactory. Regarding the High Resolution (HR) run, we considered ICs and BCs from the analysis and forecast of the the deterministic GFS forecast at 0.25 deg. resolution.*
- **'The definition of the benchmark configuration is not clearly discussed. If I look at Fig. 1 of Pichelli et al., 2017, the 1km domain (D3) does not fully cover the study area of this work, am I wrong?**
  *The Pichelli's work is on the Pò Valley, delimited by Mountain chains (Alps and Apennines), presents a complex orography as Abruzzo region. Therefore, based on the Pichelli's work we set up the deterministic operational forecast at 1km (http://magritte.aquila.infn.it/meteo/ecmwrf-2way/) over Abruzzo region. We used this operational deterministic configuration to run an 'ad hoc' deterministic forecast using 0.25 GFS, ICs and BCs, for this event and we considered it as benchmark. We clarified this point on pag. 5 at lines 21-23.*
- **In the same paper it is mentioned that the operational setup of WRF-CETEMPS is different from the one shown in Fig.1.**
  *Please refer to the previous response.*
- **Further, it seems that the high-resolution setup uses GFS later boundary forcing with different resolutions (i.e., 0.25° instead of 1° resolution) and different physical model parameterizations.'**
  *Yes, as above already mentioned, we used the best deterministic forecast produced by GFS at 0.25° and the best available GFS ensemble forecast, that is at 1 deg.. Since we used the deterministic high-resolution forecast as benchmark we decided to do not downgrade it. The use of different parameterizations, as you just stated, is driven by the different resolution. We have to use a cumulus convection parameterization at 9km which is not necessary at 1 km resolution because convection is explicitly resolved at this resolution.*
- **'I would have expected an intermediate step to reduce lateral boundary effects. In general, these aspects of the work are not clearly explained.'**
  *We agree with you. Generally, it is better to use an intermediate domain, if going down from 1° to 9km, for providing ICs and BCs to the nested domain. We performed several simulations without obtaining any significant improvement compared to the direct nesting into the GFS ICs and BCs. Therefore, based on these results, we decided to perform the ensemble forecast directly nesting the 9km to the 1° GFS ICs BCs. At this regard we added a statement on pag. 5 at lines 12-13.*

5. **'I have several remarks concerning the adopted discharge index and the discussion of the related results: - Authors consider the definition of the BDD index necessary due to the lack of discharge measurements. This contradicts the definition (Eq. 2) of the index itself, which is based on**

**discharge values! - What's the equation used to calculate the hydraulic radius as a function of the drainage area? The comparison between Fig. 10 and Fig. 9 is not intuitive. - I do not fully agree with the interpretation of Fig. 10. I see a good agreement in the timing even for those stations heavily impacted from hydropower production (i.e., Vomano and Todino). I also think that the mismatch for Pescara River could be due to some error in the observed atmospheric forcing at the local stations. That's why a more careful evaluation of the atmospheric forcing would have provided more useful insights.**

- **Authors consider the definition of the BDD index necessary due to the lack of discharge measurements. This contradicts the definition (Eq. 2) of the index itself, which is based on discharge values! - What's the equation used to calculate the hydraulic radius as a function of the drainage area?**

   *The remark is very appropriate and we thank the referee for this; we tried to summarize in one sentence two different problems, leading to a lot of confusion. The first problem deals with the difficulties to calibrate the discharge predicted by any hydrological model with observed data. Discharge observations in continuous time series are often missing, especially for small basins.*

   *A different problem is to use the predicted discharge for flood alert mapping, as it is not straightforward to establish a threshold level above which a critical event is to be expected; in addition such threshold level should be calculated for each grid point because it depends on the size of the river bed in the selected point. To overcome this second problem we tested different general definition of an alarm index and, after simulating different case studies occurring in different basins of different size, we find that a suitable definition could be the ratio between the maximum value of the predicted discharge within a given time interval and the square of hydraulic radius that is a "measure" of the river cross section for the selected point. The definition of BDD index has also a simple physical interpretation: it represents the average precipitation (more specifically the precipitation available for the runoff) drained by each grid element from the upstream basin. We explain these problems in the revised version of the paper (par. 4.1 pag 9).*

   *The BDD index is based on Eq. 2: in this equation, the used discharge value is not the measured value, but the discharge computed by the CHyM model, forced with observed rain gauges data as input.*

   *As for many other models (for a general reference see Singh and Frevert, 2002) the hydraulic radius can be approximated as a linear function of drained area. In particular $R=\beta+\gamma D^\delta$ where $\beta$, $\gamma$ and $\delta$ are empirically established and the value of $\delta$ is very close to 1. If the area is measured in $km^2$, typical values taken from literature are $\beta=0.0015$ and $\gamma=0.05$ (for a general reference please see Singh and Frevert, 2002). We added this equation in the paper (pag 10, lines 5-11).*

- **The comparison between Fig. 10 and Fig. 9 is not intuitive. - I do not fully agree with the interpretation of Fig. 10. I see a good agreement in the timing even for those stations heavily impacted from hydropower production (i.e., Vomano and Todino).**

   *We explained in details the two figures (please refer to Figures 9 and 10 of the manuscript). For what concerns figure 9 the four red triangle-shaped, thin-bounded signs indicates the relevant hydrometers where the red hydrometric threshold has been exceeded; in particular, among the involved rivers, there are Vomano, Tordino, Saline and Pescara. In figure 10, the normalized water level and BDD time series along the aforementioned rivers, for different hydrometric station grid-points, are shown.*

   *Generally, hydroelectric power installations can heavily impact the flood dynamics along a river basin, but the key parameters to be considered are various , such as the relative importance of the drained areas, the water storage capacity and the position of the reservoirs within the basin. Nevertheless, the effect highly depends on the initial reservoir filling rate, which is unknown. If the reservoirs are already full before a flood, no (gated spillway) or limited (ungated spillway) flood routing is possible (Jordan et al.,2012). Unfortunately, in Abruzzo region we are not aware of how the hydroelectric systems are managed. In this particular case, the hydroelectric power plants of Provvidenza and Piaganini are located upstream (Figs. 1 and 2, below this section), respect to the areas involved in the event, where also precipitation maxima occurred. Probably, in this case, the effect of the hydroelectric power plants is*

*negligible and this sentence is confirmed by the good agreement in the timing shown by Figure10. We added this information in the paper (pag 11, lines 8-11).*

[Figure]

Figure 1: The map represents the accumulated precipitation from 0 UTC to 12 UTC on 15th Nov 2017, as measured by the raingauges network and spatialized over the region by using the Cellular Automata-based techniques. The area enclosed in the red line is the boundary of the Tordino basin. Blue triangles indicate the position or the Provvidenza and Piaganini dams.

[Figure]

Figure 2 is a zoom of the figure 1, where the Tordino drainage network is indicated by the blue lines. Together with the dams, relevant hydrometric stations are also indicated through red pinpoints.

- **I also think that the mismatch for Pescara River could be due to some error in the observed atmospheric forcing at the local stations. That's why a more careful evaluation of the atmospheric forcing would have provided more useful insights.**
  *Please clarify what you mean by 'error in the observed atmospheric forcing'. Are you referring to the data quality at Pescara station?*

6. **For instance, authors interpret the results saying that the mismatch between "observed" and "simulated" BDD index is due to precipitation occurring only on a very small area and not capture by the model. What do you mean with "small"? The high-resolution simulations are at 1km! I suspect that you can get the same issue if you go down to 100m resolution. Again, if you do not carefully evaluate the atmospheric simulations it is difficult to provide convincing interpretation of**

the BDD index. Finally, I would also remark that authors talk about "overestimation" and "underestimation" of the BDD index using as a reference the model results driven with observed (interpolated?) precipitation. I am fine with this as long as you cross-validate local precipitation measurements with other sources of information, e.g., spatially distributed information obtained from radar retrievals.

- **For instance, authors interpret the results saying that the mismatch between "observed" and "simulated" BDD index is due to precipitation occurring only on a very small area and not capture by the model. What do you mean with "small"?**
  *The Calvano river is a very small basin (35 km^2) and is located close to Vomano final segment, southward.  The distance between the two rivers is about 2.5 km in the upper part of the Calvano's path and almost 6 km in the two mouths. Being so close, the rain spatial distribution plays a very important role: an error of even 1 km can significantly affect the forecast. Nevertheless, the Civil Protection early warning system is referred to "warning areas", rather than the single river segment or the single catchment area. For this reason, in a Decision Support System perspective, is important to assign the correct alarm state at warning area level, rather than meticulously focusing on the single catchment.*
- **Again, if you do not carefully evaluate the atmospheric simulations it is difficult to provide convincing interpretation of the BDD index.**
  *Please clarify this sentence. What do you mean by carefully evaluate atmospheric simulation? To objectively evaluate the weather forecast, as we already said, we will use skill statistical metrics, is this what you are suggesting?*
- **I am fine with this as long as you cross-validate local precipitation measurements with other sources of information, e.g., spatially distributed information obtained from radar retrievals.**
  *We compared the results with the retrieved radar precipitation, but again, being the radar precipitation a retrieved product it is affected by error as much as other observed parameters.*

7. **'I would expect the same kind of curve when I look at the black ("observed") lines in Fig. 14 and the red ones in Fig. 10, am I wrong somewhere? One of the main conclusions is that the uncertainty in the BDD index is underestimated if you do not perturb the parameters of the hydrological model. This is intuitive and this is the reason why you should take both ("atmospheric" and "hydrologic") into account. In my opinion this opportunity was missed in this work.**

- **I would expect the same kind of curve when I look at the black ("observed") lines in Fig. 14 and the red ones in Fig. 10, am I wrong somewhere?**
  *The curves appear different because of the different temporal scale. Moreover, figure 10 shows the normalized index values, whereas figure13 shows values in mm/h.*

- **One of the main conclusions is that the uncertainty in the BDD index is underestimated if you do not perturb the parameters of the hydrological model. This is intuitive and this is the reason why you should take both ("atmospheric" and "hydrologic") into account. In my opinion this opportunity was missed in this work.**
  *Based on Cloke and Pappenberger (2009) this is not an intuitive conclusion. The lack of hydrological ensemble forecast does not allow to make such statement. If you are aware of different published conclusions I would kindly ask to let us know.*

[Figure]

*Figure 1 Simulated time series (6-hourly) of precipitation at Chieti station considering the nearest grid node of global-scale driving GFS model. Ensemble mean (cyan line), ensemble members (blue lines), in black the observed time series.*

*We further discussed it in the revised version of the paper by considering also difference between ensemble mean of the Hydrological forecast and observations which appears larger than the variability among the ensemble members at most stations (Figure 13).*

**4. Page 20, lines 7 to 15. These conclusions should be reinforced. After a first, possibly fast, reading, I asked myself: "what is the relevance of the proposed method, if it merely confirms the results of the methods already in use by the civil protection agency?". Instead, the added value of the proposed method should be better discussed.**

*Again, sorry for not been clear over the mentioned statement. The alert map (Fig. 9), that we used as 'ground truth'. This map was issued in the morning of Nov 15 (i.e. the same day of the event) by the Civil Protection Agency (CPA). The map is initially built on the deterministic forecast and then updated using observations. Therefore, the methods already in use by the CPA is based on the deterministic models: the hydrological forecast forced by the meteorological forecast. This method is a very useful tool, but with short outlook. The availability of a forecast product well in advance (i.e. at least the day before of the event) would allow the CPA to issue an alert map the day before. This can be achieved by using an ensemble forecast which, though at lower resolution, produces both a forecast for a longer lead time than the deterministic, and information on the probability of the forecasted event. We accordingly improved the discussion of this point in the conclusions section of the revised version of the manuscript.*

**Technical Points.**

**1. Page 1, line 2. Correct "resolution". Rephrase "newly developed" in order to be more explicit.**
*It has been accordingly corrected and rephrased (pag. 1 line 2).*

**2. Page 1, line 5. Modify "ensemble system" in order to clarify it.**
We changed in to: A meteo-hydro off-line coupled ensemble. (*pag. 1 line 5).*

**3. Page 1, line 8. Substitute or specify "period".**
*It has been accordingly rephreased.  (pag. 1 line 8).*

**4. Page 1, lines 11 & 12. Rephrase and improve "and of the uncertainty of this flood".**
*It has been accordingly done.  (pag. 1 line 12-13).*

**5. Page 1, line 15. Substitute "one of" with "among". Correct "estimated".**

*It has been accordingly done. (pag. 1 line 16).*

**6. Page 1, line 20. Rephrase "large gradients", possibly by adding "of meteorological quantities".**
Rephrase"the cooler atmosphere and the warmer sea": which is the comparison term? cooler and warmer than what?

> *Ok, we rephrase this sentence: 'During the autumn season there is an increase of the energy available for storms (Duffourg and Ducrocq, 2011) because of large gradients of the meteorological quantities caused by the cool atmosphere and the warm sea favoring heat and moisture fluxes.' (pag. 1 line 21-23).*

**7. Page 1, line 23. Rephrase "Recent decades".**
*It has been accordingly done. (pag. 2 line 2).*

**8. Page 2, line 2; page 25, line 24. Check the publication date of Van den Besselaar et al. (2011).**
*The reference has been accordingly corrected. (pag. 28 line 35 and pag. 2 line 3).*

**9. page 2, lines 3 & 4. Rephrase "a warmer atmosphere and a greater amount of water vapor": which is thecomparison term?**
*It has been accordingly done. (pag. 2 line 4).*

**10. Page 2, line 4. Check spelling of "Willet".**
*The spelling has been accordingly corrected. (pag. 29 line 7).*

**11. Page 2, line 6. Substitute "to" with "on".**
*It has been accordingly done. (pag. 2 line 8).*

**12. Page 2, line 9. Is "occur" the proper word? May be, "are expected"?**
*We substituted 'occur' with 'is expected'. (pag. 2 line 11).*

**13. Page 2, line 10. EU Flood Directive (2007) is referenced wth a different "author name" in the referencelist. Add "," after "2007)". Provide references for "Recent studies".**
*We added a few references. (pag. 2 line 12-13).*

**14. Page 2, line 12. Is "lead time" the right expression?**
*We change lead time into outlook. (pag. 2 line 15).*

**15. Page 2, line 15. "Assuming an appropriate hydrological model formulation": this is not a weak assumption,this should be discussed more accurately.**
*We added a statement pag 2, lines 18-19.*

**16. Page 2, line 16. Correct "scales".**
*It has been accordingly done. (pag. 2 line 20).*

**17. Page 2, line 23. Substitute "on" with "in".**
*It has been accordingly done. (pag. 2 line 27).*

**18. Page 2, line 25; page 8, lines 13 & 17; page 19, line 9. Add "with" before "respect".**
*It has been accordingly done.*

**19. Page 2, line 26. "their added values belong" or "their added value belongs".**
*It has been accordingly done. (pag. 2 line 29).*

**20. Page 3, line 10. Erase "ing" from "fostering".**
*It has been accordingly done. (pag. 3 line 15).*

**21. Page 3, line 11. Substitute "placed in", possibly with "associated to".**
*It has been accordingly done. (pag. 3 line 15).*

**22. Page 3, line 17. The GFS acronym has been introduced without explanation.**

*The GFS acronym has been introduced. (pag. 3 line 23).*

**23. Page 3, lines 23 to 25. Rephrase the sentence "The results... extreme events".**
*We accordingly modified the sentence. (pag. 3 line 31-35).*

**24. Page 3, line 26. Substitute "as it has been discussed by", possibly with "on the basis of the results of".This topic should be discussed in more detail.**
*We changed the sentence and we added more details in the discussion. (pag. 4 line 10-15).*

**25. Page 4, line 4. Add "," after "trough".**
*The comma has been added.*

**26. Page 4, lines 9 & 10. Rephrase "The thermal advection... at the upper ones": thermal advection is aphysical process and I am afraid that is not correct to associate this expression to adjectives like "warm"and "cold".**
*We accordingly changed the sentence. (pag. 4 line 23-24).*

**27. Page 4, line 11. Please replace "13 November 2017" with a more precise definition of the timing of thetwo phases.**
*It has been accordingly done. (pag. 4 line 25-28).*

**28. Page 4, line 12. Please rephrase "not shown". Do you mean, not represented in the figures or in some ofthem? Or not discussed in the paper?**
*We meant not presented in the figure and not discussed. We have accordingly changed the sentence.*

**29. Page 4, line 18. Substitute "because of", possibly with "as evidenced by".**
*It has been accordingly done. (pag. 4 line 33).*

**30. Page 4, line 27. Substitute "Similarly to", possibly with "Following" or "In accordance with". Erase ","after "study".**
*It has been accordingly done. (pag. 6 line 8-9).*

**31. Page 8, lines 13 to 15. In the statistical literature, this is simply called the "ensemble standard deviation".**
*It has been accordingly done.*

**32. Page 8, lines 19 & 18. In the statistical literature, this is simply known as the "coefficient of variation".**
*It has been accordingly done.*

**33. Page 8, line 21. Add "it" before "has been".**
*It has been accordingly done.*

**34. Page 8, line 28. Substitute "this latter"; it is not clear.**
*We substitute "this latter" with the "control simulation".*

**35. Page 10, line 25. Check spelling of "Lighthill".**
*It has been accordingly modified.*

**36. Page 10, line 32. The word "based" is repeated twice at short distance.**
*We accordingly changes the verb.*

**37. Page 13, line 3. Erase "," after "comparison". Erase "even".**
*It has been accordingly removed.*

**38. Page 13, line 4. Rephrase "hourly data in m".**
*It has been accordingly done. (pag. 10 line 24).*

**39. Page 20, line 11. Correct "initialized".**

*It has been accordingly corrected.*

**40. Page 20, lines 19 & 20. Rephrase "as a complementary tools by using both".**
*We have rephrased the sentence. (pag. 15 line 14-28).*

**41. Page 23, line 20. Check "Coauthors".**
*The reference has been corrected.*

**42. Page 24, lines 11 to 14. These references are cited with a different, abbreviated author's name in the text.**
*The references have been corrected.*

**43. Figure 1. Add contour spacing for the two represented quantities in the figure captions. Expand theacronym "mslp". Black lines are hardly visible; I think that they would be more visible if drawn in yellow.**
*We have accordingly corrected the Figure 1 except for the line color that is a default color of the EumetRain tool and unfortunately not changeable by the users.*

**44. Figure 2. I am afraid that it will be very difficult to read the legend in the printed version of the paper. In the caption, add a space in "24 hours".**
*It has been accordingly done.*

**45. Figure 6. I think that a better choice of the colours could help to examine the plots. In particular,inverting the colours for the Ensemble members and the Ensemble mean could help to visualize the meanmore easily.**
*Changing Fig. 6 entails also changing Fig. 13 to be coherent. It unfortunately takes some time; therefore we would ask for the possibility of correcting it in the next revisions round or for the publishing version of the paper.*

**46. Table 1. The first column can be erased because it is not informative: it has the same value for all thesimulation types. I think that it could be useful to add a column to assign a code to each simulation type.**
*We agree, we assigned a code to each simulation and we corrected the text.*

[revised manuscript text omitted]

---

## Referee Report (RR1)

Review of the revised manuscript HESS-2019-223
"Regional ensemble forecast for early warning system over small
Apennine catchments on Central Italy"
by Ferretti R, Lombardi A, Tomassetti B, Sangelantoni T, Colaiuda V,
Mazzarella V, Maiello I, Verdecchia M, Redaelli G

25 November 2019

**1  General comments**

The manuscript has been revised following the reviewers' comments. Some answers can be considered satisfactory and the manuscript has been generally improved. However, some scientific flaws are still present, as listed below under the specific comments, and they should be fixed before accepting the manuscript for publication.

**2  Specific comments**

1. The statistical analysis of the results has been slightly reinforced by introducing the analysis of ROC. However, I am worried about some results shown in Figure 7. Please, excuse me, if I am missing some details and I misunderstand the result, but I think there is some mistake in data processing. Some basic remarks follow.

   (a) ROC curves should be monotonically non-decreasing, shouldn't they? The probability of detection and the false alarm rate for a threshold (e.g., 10 mm) are always smaller than those for lower thresholds (e.g., 5 mm), aren't they? This is not the case for the blue curve in Figures 7a and 7b and for the red curve in Figure 7b.

   (b) How can be computed the area under the blue curves, which have multiple values on the $y$ axis for a given interval on the $x$ axis?

   (c) Why are the data of the threshold 15 mm not shown for the red and pink curves?

2. I am sorry, but I did not put enough attention to equation (1) in the previous review. This equation is evidently wrong. The following remarks should also be considered.

   (a) Notice that $freq = (t_{mem+1} + 1)/2$, because $\sum_{j=1}^{n} j = n \times (n+1)/2$.

   (b) Why is the sum extended to $t_{mem+1}$ if the number of members of the ensemble is $t_{mem}$? Is $t_{mem+1}$ the control member?

3. Page 7, lines 28 to 30. The sentence "all the time series show a variability among ensemble members much smaller than the difference between ensemble mean and observations at the maximum of the rainfall" implies that the variability among the ensemble members can be used to estimate the uncertainty among the predictions, but it does not provide an accurate estimate of the spread between the observed and the predicted quantities. Unfortunately, I think that this is a very weak point of the manuscript.

4. In their answer, the Authors claim that "there are not yet operational system based on coupled regional meteorological ensemble and hydro ensemble in Italy", but this is not enough to support the innovation content of a scientific paper in an international high-quality journal.

**3  Technical comments**

1. Page 5, lines 1 to 3. Acknowledge in the text the data source for SRT and rain gauges. Should not SRT be substituted with STR (Surface total rainfall)?

2. Page 5, line 13. In which sense the following configuration provides the best results?

3. Page 8, line 11. The ROC method was proposed much earlier than the book by Jolliffe and Stephenson (2012), which, therefore, could not be the best reference.

4. Page 8, line 15. I expect that radar data are spatially distributed over the area and do not need interpolation. Which interpolation method is used for the rain gauges data?

---

## Author Response (AR2)

Review of the revised manuscript HESS-2019-223 "Regional ensemble forecast for early warning system over small Apennine catchments on Central Italy" by Ferretti R, Lombardi A, Tomassetti B, Sangelantoni T, Colaiuda V, Mazzarella V, Maiello I, Verdecchia M, Redaelli G

28 February 2020

*Specific comments*

1. *The statistical analysis of the results has been slightly reinforced by introducing the analysis of ROC. However, I am worried about some results shown in Figure 7. Please, excuse me, if I am missing some details and I misunderstand the result, but I think there is some mistake in data processing. Some basic remarks follow.*

   (a) *ROC curves should be monotonically non-decreasing, shouldn't they? The probability of detection and the false alarm rate for a threshold (e.g., 10 mm) are always smaller than those for lower thresholds (e.g., 5 mm), aren't they? This is not the case for the blue curve in Figures 7a and 7b and for the red curve in Figure 7b.*

   The ROC curves have been recalculated due to a problem with the data files. The new results fix the evolution of ROC curves for ensemble forecast, except for the high resolution. This still shows some inconsistency between 10 mm/3h and 5 mm/3h thresholds because of high spatial variability of precipitation over a complex orography terrain. We point out that the aim of this paper is to present a meteo-hydro ensemble and eventually to validate the ensemble without performing a comparison between the HR and the ensemble. Therefore, we decide not to show the ROC curves for the HR simulations, which are not necessary for an evaluation of the ensemble but to evaluate the ensemble and the CNTR simulation. We rewrite the related manuscript section (pag. 9, lines 3-8)

   (b) *How can be computed the area under the blue curves, which have multiple values on the y axis for a given interval on the x axis?*

   The new curves do not show this problem. Therefore, the Area Under the Curve (AUC) can be computed exactly. A table with the values of the AUC, calculated for two-time steps, has been added to the paper (pag. 9, table 1)

   (c) *Why are the data of the threshold 15 mm not shown for the red and pink curves?*

   The 3 hourly accumulated rainfall does not reach the threshold value, that is why we did not show the 15mm related curve

2. *I am sorry, but I did not put enough attention to equation (1) in the previous review. This equation is evidently wrong. The following remarks should also be considered.*

   (a) *Notice that freq = $(t_{mem+1} + 1)/2$, because $\sum_{j=1}^{n} j = n \times n(n + 1)/2$*

   (b) *Why is the sum extended to $t_{mem+1}$ if the number of members of the ensemble is $t_{mem}$? Is $t_{mem+1}$ the control member?*

   a) We are deeply sorry, but we made a mistake in writing the equation and we did not notice it in reviewing the paper. Thanks for pointing it out! We correct equation 1 in the paper (from pag 6, line 30 to pag.7 line 5).

   b) Yes, the summation is plus 1 because the control member is also added. We added a statement in the paper

3. *Page 7, lines 28 to 30. The sentence "all the time series show a variability among ensemble members much smaller than the difference between ensemble mean and observations at the maximum of the rainfall" implies that the variability among the ensemble members can be used to estimate the uncertainty among the predictions, but it does not provide an accurate estimate of the spread between the observed and the predicted quantities. Unfortunately, I think that this is a very weak point of the manuscript.*

   We performed the computation of the rank histogram (Fig. 1) to verify the variability of the ensemble respect to the observed values and we found that there is too little spread as you suggested, but no bias. We would expect to have a uniform distribution for a properly calibrated ensemble even if this is not a sufficient criterion for thoroughly assessing the ensemble reliability (Hamill, 2001). Moreover, if a single step rank histogram is considered a closer distribution to the uniform is found, except for member 21 (Fig. 2).

[Figure]

Figure 1

[Figure]

Figure 2

In this regard a comment has been added on pag. 8 lines 16-21.

We want to stress out that this paper aims to present a new meteo-hydro ensemble forecast, where the regional meteorological ensemble represents the driving field of the hydrological ensemble, which on the contrary shows a larger variability. In any case, being the meteorological ensemble an intermediate step to build the hydrological one, which is the final aim of this work, a conclusive assessment on the ensemble reliability can be expressed according to the benefits on the hydrological forecast, if any.

4. *In their answer, the Authors claim that "there are not yet operational system based on coupled regional meteorological ensemble and hydro ensemble in Italy", but this is not enough to support the innovation content of a scientific paper in an international high-quality journal.*

Probably, we did not state clearly the following point: actually, this paper is not on the regional meteorological ensemble, but on the development of a high-resolution hydrological ensemble on a complex topography, forced by a regional meteorological ensemble. The newly developed hydrological ensemble applied to the small Apennine catchments and the probability index (BDD$_{prob}$) specifically developed for assessing the uncertainty of the flood risk, are part of the innovation content of the paper. We added a few statements to make it clearer (pag 1 lines 4-8; pag. 4, lines 1-2; pag.15, lines 31-33; pag. 16, lines 1-5)

Moreover, to the aim of preventing misunderstandings in the purpose of the paper, we also changed the title to: "Regional meteo-hydro ensemble forecast for early warning system over small Apennine catchments on Central Italy"

**3    Technical comments**

1. *Page 5, lines 1 to 3. Acknowledge in the text the data source for SRT and rain gauges. Should not SRT be substituted with STR (Surface total rainfall)?*

On the DEWETRA platform is defined as SRT and we prefer to keep as it is

2. *Page 5, line 13. In which sense the following configuration provides the best results?*

The best results in term of precipitation forecast. We clarified this in the paper.

3. *Page 8, line 11. The ROC method was proposed much earlier than the book by Jolliffe and Stephenson (2012), which, therefore, could not be the best reference.*

Thanks for the suggestion, we changed the references in the paper for the use of ROC in atmospheric science (pag 21, lines 23-24).

4. *Page 8, line 15. I expect that radar data are spatially distributed over the area and do not need interpolation. Which interpolation method is used for the rain gauges data?*

The data are interpolated to the model grid, this has been done also for the radar data. Inverse Distance Weighting (IDW) Conservative method (Jones, PW. 1999) has been used for this purpose.

References:

Hamill, 2001: Interpretation of Rank Histograms for Verifying Ensemble Forecasts. Mon. Wea. Rev., https://doi.org/10.1175/1520-0493(2001)129%3C0550:IORHFV%3E2.0.CO;2

Jones, PW. 1999: First- and Second-Order Conservative Remapping Schemes for Grids in Spherical Coordinates Mon. Wea. Rev. , 127, 2204-2210

---

## Author Response (AR3)

Dear Editor,
we followed your suggestion changing the abstract.
Please find the change marked in red  on the pdf file.
Thanks
Best Regards,
Rossella ferretti